# Genetic therapy in a mitochondrial disease model suggests a critical role for liver dysfunction in mortality

**Ankit Sabharwal[1†], Mark D Wishman[1†], Roberto Lopez Cervera[1†], MaKayla R Serres[1], Jennifer L Anderson[2], Shannon R Holmberg[1], Bibekananda Kar[1], Anthony J Treichel[1], Noriko Ichino[1], Weibin Liu[1,3], Jingchun Yang[1,3], Yonghe Ding[1,3], Yun Deng[1,3], Jean M Lacey[4], William J Laxen[4], Perry R Loken[4], Devin Oglesbee[4], Steven A Farber[2], Karl J Clark[1], Xiaolei Xu[1,3], Stephen C Ekker[1]\***

[1]Department of Biochemistry and Molecular Biology, Mayo Clinic College of Medicine, Rochester, United States; [2]Department of Embryology, Carnegie Institution for Science, Baltimore, United States; [3]Division of Cardiovascular Diseases, Department of Medicine, Mayo Clinic College of Medicine, Rochester, United States; [4]Biochemical Genetics Laboratory, Department of Laboratory Medicine and Pathology, Mayo Clinic College of Medicine, Rochester, United States

**\*For correspondence:**
ekker.stephen@mayo.edu

[†]These authors contributed equally to this work

**Competing interest:** The authors declare that no competing interests exist.

**Abstract** The clinical and largely unpredictable heterogeneity of phenotypes in patients with mitochondrial disorders demonstrates the ongoing challenges in the understanding of this semi-autonomous organelle in biology and disease. Previously, we used the gene-breaking transposon to create 1200 transgenic zebrafish strains tagging protein-coding genes (Ichino et al., 2020), including the *lrpprc* locus. Here, we present and characterize a new genetic revertible animal model that recapitulates components of Leigh Syndrome French Canadian Type (LSFC), a mitochondrial disorder that includes diagnostic liver dysfunction. LSFC is caused by allelic variations in the *LRPPRC* gene, involved in mitochondrial mRNA polyadenylation and translation. *lrpprc* zebrafish homozygous mutants displayed biochemical and mitochondrial phenotypes similar to clinical manifestations observed in patients, including dysfunction in lipid homeostasis. We were able to rescue these phenotypes in the disease model using a liver-specific genetic model therapy, functionally demonstrating a previously under-recognized critical role for the liver in the pathophysiology of this disease.

## Editor's evaluation

This study presents an interesting discovery that reversion of the mutation in the liver largely rescues the lethality in a zebrafish model of Leigh Syndrome, French Canadian Type (LSFC). The evidence is strong as 90% of the liver revertants survive past the terminal age. This manuscript is of interest to scientists in the field of mitochondrial biology and pathology.

## Introduction

The mitochondrion is a complex and essential organelle whose dysfunction is linked to a panoply of diverse human pathologies and diseases. The understanding of the traditional role of mitochondria as the powerhouses of the cell has evolved as its multiple roles in cellular and physiological homeostasis have been functionally enumerated far beyond its commonly known role in energy production via the

electron transport chain (ETC) (*Spinelli and Haigis, 2018*). All these functions rely on a coordinated cross-talk between the nuclear genome and its mitochondrial counterpart (*Vafai and Mootha, 2012*). The widespread activity of mitochondria provides ample opportunities for mitochondrial dysfunction to play a role in human disease, but current understanding does not provide specificity in differential biological function or capacity in accurately predicting pathogenic details suitable for therapeutic development.

Here we focus on Leigh Syndrome, French Canadian Type (LSFC), an autosomal recessive mitochondrial disease with onset in infancy that manifests with diagnostic liver dysfunction (*Morin et al., 1993*). The most common allelic variant is due to an A354V transition in exon 9 of the leucine-rich pentatricopeptide repeat-containing motif (*LRPPRC*) protein (*Mootha et al., 2003*). Patients with mutations in the *LRPPRC* gene experience a diverse array of clinical features centered around cytochrome c oxidase (COX) deficiency. These can include early onset of hepatic micro vesicular steatosis and chronic lactic acidosis in neonates. Individuals born with LSFC have an average lifespan of less than 5 years of age with most succumbing to a series of extreme, acute metabolic crises. Patients that make it past the early life crises show a lessened disease state characterized by hypotonia, language and mobility deficits and delays, and muscle weakness (*Morin et al., 1993*; *Debray et al., 2011*).

LRPPRC belongs to the pentatricopeptide repeat (PPR) containing motif family of proteins. The LRPPRC protein in humans has been reported to have 33 predicted PPR domains and is promiscuous, preferentially binding to different mitochondrial RNAs (*Spåhr et al., 2016*). *LRPPRC* plays role in the regulation of both nuclear and mitochondrial RNA expression at the transcriptional and post-transcriptional levels (*Liu and McKeehan, 2002*; *Mili and Piñol-Roma, 2003*). Loss of functional *LRPPRC* in human fibroblasts, *C. elegans,* and mice revealed decreased stability of mitochondrial mRNAs, defects in mitochondrial biogenesis, decreased complex 1 and cytochrome c oxidase activity, and dysregulated mitochondrial translation (*Cuillerier et al., 2017*; *Gohil et al., 2010*; *Xu et al., 2012*; *Köhler et al., 2015*). Given that onset of LSFC is marked by the neurological and metabolic crisis (*Morin et al., 1993*) and the liver is the major metabolic factory of the organism, therefore, the role of liver impairment in LSFC represents a unique aspect of mitochondrial disease unseen in traditional Leigh syndrome. Mice harboring liver-specific inactivation of *Lrpprc* mirrored similar clinical manifestations marked by mitochondrial hepatopathy, growth delay, and reduced fatty acid oxidation compared to controls (*Cuillerier et al., 2017*). Recently, a lipidomic profile study on LSFC patients highlighted a novel role of this protein in the peroxisomal lipid metabolism (*Ruiz et al., 2019*). The current diagnosis of LSFC hinges on measuring lactate levels in the blood and brain, COX activity in patient fibroblasts, and sequencing of the *LRPPRC* gene for mutations (*Debray et al., 2011*). The lack of a cure or effective therapies for LSFC patients signifies that current clinicians must shift their focus to relieving and controlling symptoms. Focus on dietary restrictions and lifestyle changes, including exercise and infection risk management, are made to reduce the physiological and metabolic stress on the patient. To build upon a heuristic paradigm for therapeutic interventions for these classes of disorders, we need to establish disease models that serve as a suitable substrate to recapitulate the key functional tissue-specific pathology of the disease.

Zebrafish have been employed as a powerful potential vertebrate model organism to study human mitochondrial disorders (*Broughton et al., 2001*; *Steele et al., 2014*; *Sabharwal et al., 2019a*). Previously, we have reported a compendium of approximately 1200 zebrafish transgenic lines created using the gene-breaking transposon approach out of which 204 were clonally validated and molecularly characterized (*Ichino et al., 2020*). The GBT construct is incorporated into the intron of an endogenous gene via Tol2 transposase and uses a splice acceptor site to create a fusion protein between the native transcript and red fluorescent protein (*mRFP*) gene that contains a transcription termination site. When inserted intergenically, this results in the translation of a truncated endogenous gene product fused to *mRFP*. This enables spatiotemporal tracking of the trapped gene's natural protein expression in real-time. This insertion is reversible due to two *loxP* sites located near each terminal repeat that enables the removal of the GBT cassette using *Cre* recombinase. The revertible nature of this construct allows for the reversion of the mutant back to wild-type alleles to allow for tissue-specific or ubiquitous rescue of the mutated gene (*Ichino et al., 2020*; *Clark et al., 2011*). The previous work (*Ichino et al., 2020*) reported over thirty new potential new models of human disease; we report here the characterization and detailed investigation of one, the mitochondrial protein-encoding locus *lrpprc*.

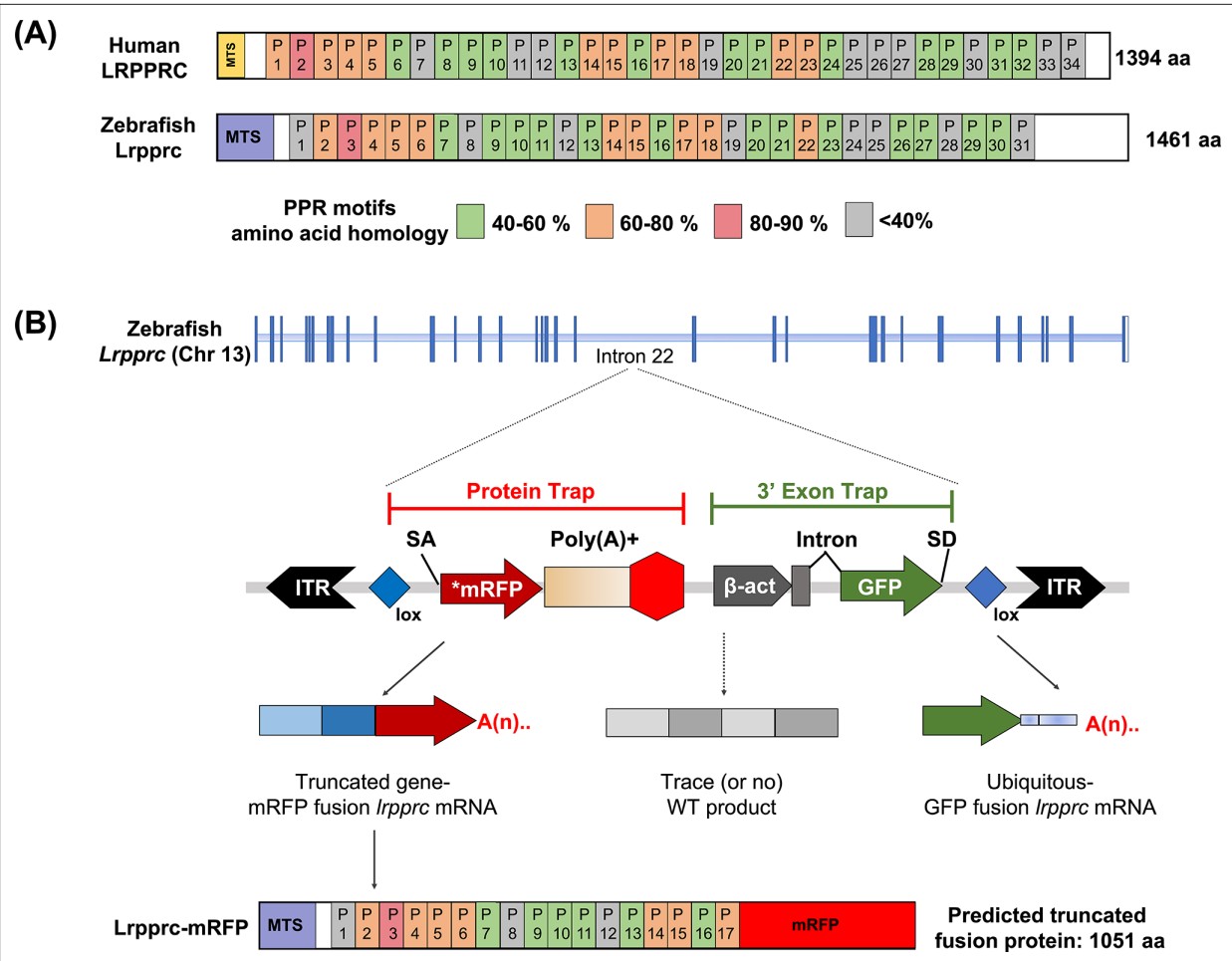

**Figure 1.** GBT mutagenesis generates a novel zebrafish model of LSFC. (**A**) Schematic of human and zebrafish LRPPRC proteins with highlighted PPR domains (denoted by P). (**B**) Schematic of the integration event of GBT vector RP2.1 with 5' protein trap and 3' exon trap cassettes. The RP2.1 cassette was integrated into the intronic region 22 of the *lrpprc* genomic locus on chromosome 13. ITR, inverted terminal repeat; SA, *loxP*; *Cre* recombinase recognition sequence, splice acceptor; *mRFP*' AUG-less mRFP sequence; poly (**A**) +, polyadenylation signal; red-octagon, extra transcriptional terminator and putative border element; β-act, carp beta-actin enhancer, SD, splice donor.

We describe here the first revertible zebrafish mutant with a single disruptive insertion in the *lrpprc* gene to model LSFC. *lrpprc* homozygous mutants mimic many hallmarks observed in patients such as early lethality, defective muscle development, and decreases in mitochondrial transcript levels. In addition, altered dietary lipid metabolism along with mitochondrial dysmorphology was also observed in these mutants. A key outcome of this study is the reversion of the LSFC-associated phenotype and survival using liver-specific *Cre* recombinase in the homozygous *lrpprc* mutants, demonstrating a critical role of this organ in the pathogenesis of disease in this animal model. These results contribute to a novel paradigm for the identification of therapeutic approaches for LSFC and other related disorders.

## Results

### Spatiotemporal expression dynamics of GBT0235 tagged by the *in vivo* protein trapping

Molecular analyses revealed that the *in vivo* protein trap line GBT0235 contained a single RP2.1 integration event within intron 22 of the *lrpprc* gene on chromosome 13 (*Figure 1A–B*). The RP2.1 GBT protein trap cassette overrode the transcriptional splicing machinery of the endogenous *lrpprc* gene, creating an in-frame fusion between the upstream endogenous exons and the start codon-deficient *mRFP* reporter sequence. This mRFP-fusion resulted in a translation product that was predicted to

be truncated from 1461 amino acids to 817 amino acids (danRer11). The direct protein comparison between zebrafish and human Lrpprc predicted proteins revealed a complex set of putative PPR motifs, many of which had different levels of homology (*Figure 1A*).

## Structure-based analysis reveals Lrpprc-binding sites in the mitochondrial transcriptome

PPR proteins contain tandem arrays of degenerate ~35-amino acids motifs that fold into a hairpin of antiparallel alpha-helices (*Barkan et al., 2012*; *Yin et al., 2013*). PPR proteins bind to their target RNAs in a modular and base-specific fashion (*Barkan et al., 2012*). The 5th and 35th amino acids of each PPR motif determine the target nucleotide, and the ordering of the repeat motif elements in the respective PPR proteins can function as determinants to assign sequence binding specificity (*Yin et al., 2013*). To investigate the functional homology for PPR motifs in Lrpprc orthologs from humans and zebrafish, we first wanted to identify and annotate the PPR motifs. Since PPR motifs fold into a distinct structure of antiparallel alpha-helices, we were interested to interrogate the structure of the Lrpprc protein to estimate the occurrence of such helices across its tertiary structure. As no traditional structures for these protein orthologs were available, we derived predicted models using the AI-based protein structure prediction tool, RoseTTAFold. These models were used to identify the PPR motifs throughout the human and zebrafish Lrpprc proteins and to evaluate any structural similarities between the orthologs. Interestingly, many of the resulting derived PPR motifs in human LRPPRC were observed to be shorter than the canonical 35 amino acid PPR consensus motifs (*Figure 2*). However, the human and zebrafish Lrpprc protein structures revealed that the signature PPR motif helix-turn-helix repeats were packed in a rigid conformation (*Figure 2A–B*). We manually annotated the occurrence of the hairpin of α-helices (PPR motifs) throughout the human and zebrafish Lrpprc models and numbered each starting from the *N*-termini of the proteins. Each hairpin of the α-helix is believed to bind to a single nucleotide target. This analysis identified 34 and 31 PPR motifs in the human and zebrafish Lrpprc orthologs, respectively (*Figure 2—figure supplements 1 and 2*).

Analysis of these structures identified a series of standard PPR motifs and their resulting likely RNA-targeting amino acid side chains for sequence-specific interactions. Some PPR motifs of the human and zebrafish Lrpprc proteins were also noted to be less regular in form. For example, some motifs did not have obvious typical amino acids at the 5th and 35th positions to confer the nucleotide specificity and were longer or shorter than typical canonical PPR motifs. The resulting inferred consensus sequences (*Figure 2—figure supplements 1 and 2*) represent candidate RNA binding targets for these proteins.

To predict the potential transcripts bound by Lrpprc proteins, we took advantage of the recent development in executing the FIMO program to search the potential targets for several plant PPR proteins (*Takenaka et al., 2013*; *Hammani et al., 2016*). The predicted putative targets for human and zebrafish Lrpprc (*Figure 2—figure supplements 1 and 2*) were used to query the complete human and zebrafish mitochondrial transcriptomes, respectively. We identified several mitochondrial-encoded genes as targets using the FIMO motif search tool of the MEME Suite, and the top twenty hits with high confidence scores are listed in *Supplementary file 1B and C*, for humans and zebrafish, respectively. In addition, the structure of zebrafish Lrpprc protein revealed that the C-terminus half, which was deleted in the zebrafish GBT mutant allele, consisted of multiple PPR motifs for RNA-binding (*Figure 2C* and *Figure 2—figure supplement 2*).

## *lrpprc*<sup>GBT0235/+</sup> mutants exhibit mitochondrial localization of Lrpprc-mRFP protein

RFP expression analysis of GBT0235 during zebrafish embryonic development revealed the onset of reporter expression as early as one-cell stage that could be traced until late larval stages. For instance, animals homozygous for the GBT0235 allele (hereafter *lrpprc*<sup>GBT0235/GBT0235</sup>) exhibit ubiquitous RFP expression throughout the body at 6 dpf with strong expression in the liver, gut, and muscles (*Figure 3A*). Endogenous *lrpprc* transcript levels were negligible in *lrpprc*<sup>GBT0235/GBT0235</sup> as compared to *lrpprc*<sup>+/+</sup>, indicating a nearly complete knockdown (*Figure 3B*).

To investigate the subcellular localization of the mRFP fusion protein, we crossed *lrpprc*<sup>GBT0235/+</sup> to *Tg(MLS:EGFP)* zebrafish. The caudal fin presents a unique advantage in studying the mitochondrial sub-cellular network *in vivo* as it is as few as two cells thick and myocytes possess a rich network of

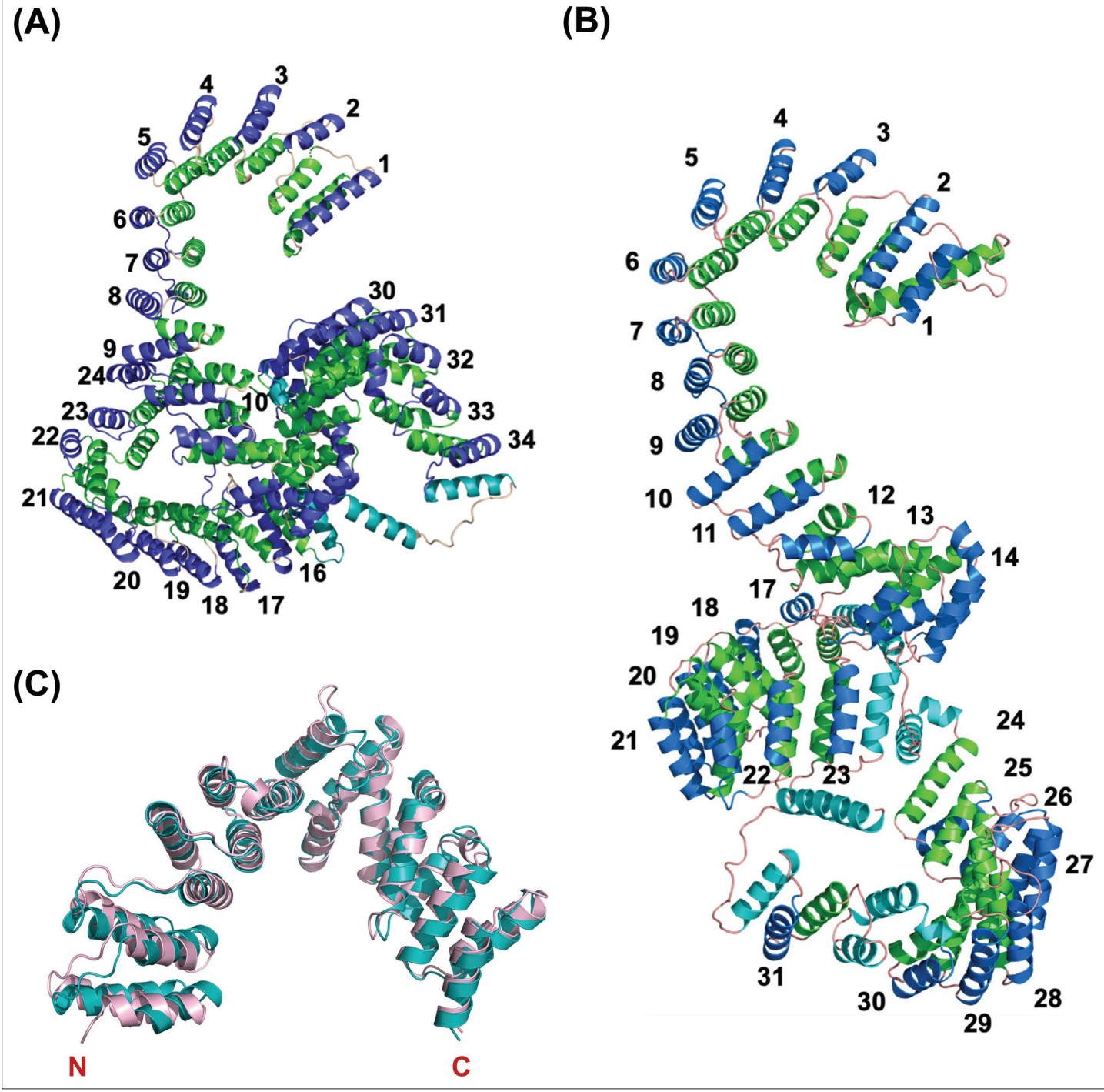

**Figure 2.** Overall predicted structure of human and zebrafish LRPPRC and its C-terminus comparison with human LRPPRC. (**A**) The predicted structure of mature human LRPPRC comprises 34 likely PPR repeats. (**B**) Overall predicted structure of mature zebrafish Lrpprc comprises 31 likely PPR repeats. The two helices within each repeat are colored green and blue. The predicted PPR motifs are numbered from the N-terminal of the protein. The functionally undefined regions and C-terminal helix were colored cyan. (**C**) Structural superimposition of the C-terminus part (SEC1 domain) of human and zebrafish Lrpprc. The light pink and teal colors represent human and zebrafish Lrpprc respectively.

The online version of this article includes the following figure supplement(s) for figure 2:

**Figure supplement 1.** Domain organization of human LRPPRC predicted from structural and sequence analysis.

**Figure supplement 2.** Domain organization of zebrafish Lrpprc predicted from structural and sequence analysis.

**Figure supplement 3.** Sequence alignment of SEC1 domain of LRPPRC from different model organisms.

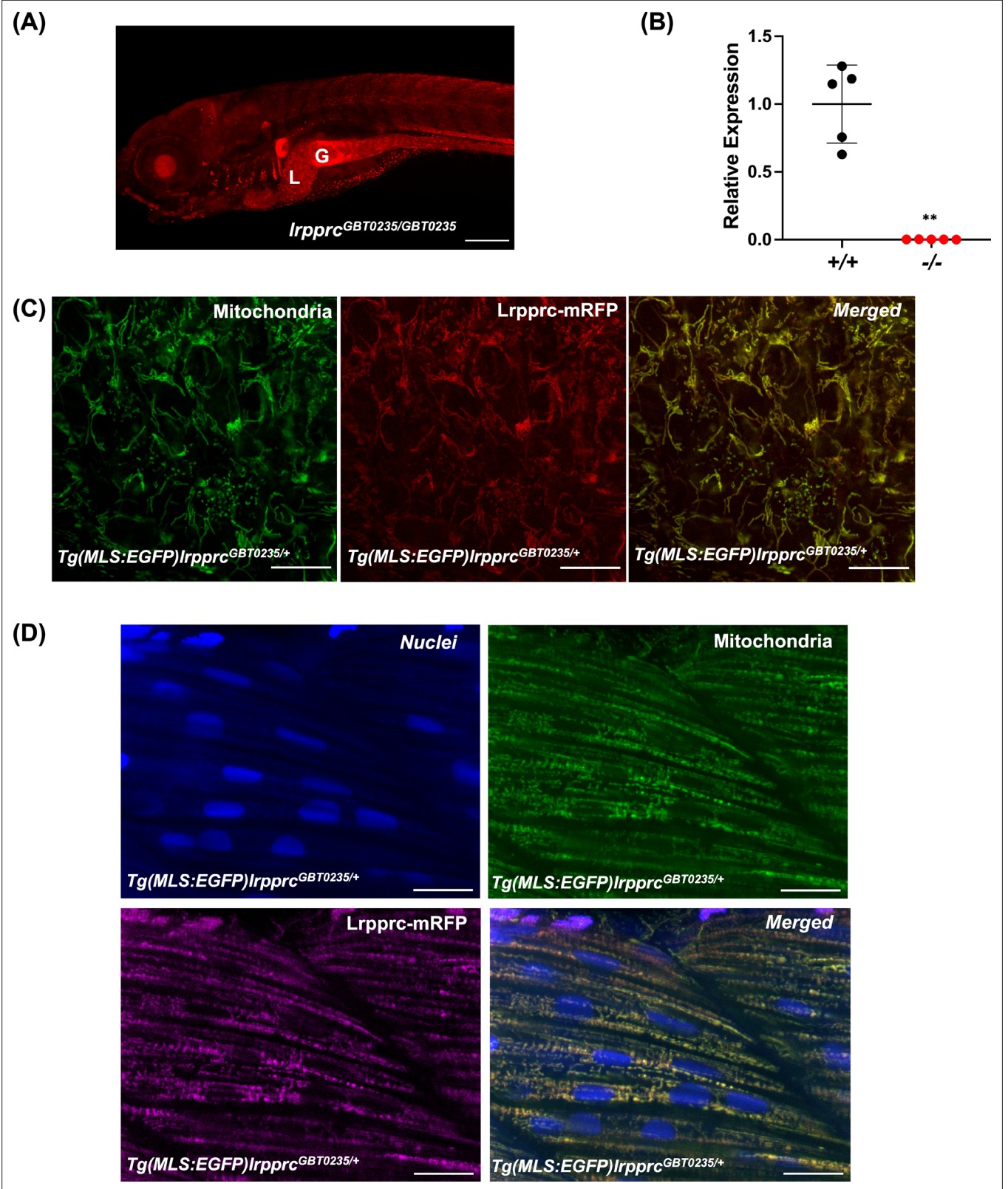

**Figure 3.** Spatiotemporal expression of Lrpprc-mRFP in GBT0235 mutants. (**A**) Representative images of 6 dpf *lrpprc*$^{GBT0235/GBT0235}$ with RFP expression in the liver and gut (magnification-5×; scale-bar: 200 μm). (**B**) Relative expression of *lrpprc* transcript in *lrpprc*$^{+/+}$ and homozygous mutant larvae, *lrpprc*$^{-/-}$ (*P*-value = 0.0015). p-value was determined using the unpaired t-test. Each data point represents a biological replicate (N=5) (*Figure 3—source data 1*). (**C**) Mitochondrial network marked by EGFP in the caudal fin was observed to be overlapping with truncated Lrpprc:mRFP fusion protein in 2 dpf

*Figure 3 continued on next page*

*Figure 3 continued*

*Tg(MLS:EGFP)lrpprc^GBT0235/+* embryo (scale bar: 15 µm). (**D**) RFP from truncated Lrpprc:mRFP fusion protein was observed to be overlapped with EGFP in mitochondria present in myocytes of skeletal muscle region in 2 dpf *Tg(MLS:EGFP)lrpprc^GBT0235/+* embryo injected with NLS:TagBFP RNA. Nuclei were marked by TagBFP protein (scale-bar: 15 µm).

The online version of this article includes the following source data and figure supplement(s) for figure 3:

**Source data 1.** Numeric data for the relative expression of *lrpprc* transcript in *lrpprc^+/+* and homozygous mutant larvae, *lrpprc^-/-*.

**Figure supplement 1.** Lrpprc-mRFP localizes to the mitochondria in the zebrafish mutants.

mitochondria that can be imaged easily. Images were acquired of the caudal fin at 2 dpf (*Figure 3C*) and of myocytes at 2 dpf (*Figure 3D*) and 4 dpf (*Figure 3—figure supplement 1*). Across these tissue systems, the reticular mitochondrial network was marked by the GFP (*Tg(MLS:EGFP)*), and the expression pattern of the Lrpprc was marked by mRFP in the mitochondria. Composite images revealed an overlap between the GFP and Lrpprc-mRFP fusion protein in the caudal fin of the zebrafish larvae (*Figure 3C*). To image the myocytes and observe the sub-cellular distribution, single-cell zebrafish embryos obtained from the outcross of *lrpprc^GBT0235/+* and *Tg(MLS:EGFP)* were injected with NLS:TagBFP (Nuclear localization signal:Tag blue fluorescent protein) RNA. Nuclei of myocytes were marked by BFP whereas an overlap was observed between the GFP and Lrpprc-mRFP fusion protein in the muscle region (*Figure 3D*).

## *lrpprc^GBT0235/GBT0235* mutants develop hallmarks of LSFC

LSFC patients exhibit several hallmarks such as metabolic/acidotic crisis followed by death within the first 2 years of life. To assess the survivability of our GBT mutants, we examined the larvae during the first 12 days of development and obtained a survival curve for *lrpprc^+/+*, *lrpprc^GBT0235/+*, and *lrpprc^GBT0235/GBT0235*. *lrpprc^GBT0235/GBT0235* mutants had a similar survival percentage to that of *lrpprc^+/+* and *lrpprc^GBT0235/+* through 6 dpf, followed by marked mortality leading to 100% lethality at the end of the follow-up on 12 dpf (*Figure 4A*). In contrast, the survival trend was similar between the heterozygous mutants and the wild-type animals. The mortality rate was thus strongly affected by the *lrpprc^GBT0235/GBT0235* genotype. To assess the impaired function of the *lrpprc* gene, an investigation of the transcript levels of mitochondrial-encoded genes revealed a significant drop-off in their expression in the *lrpprc^GBT0235/GBT0235* mutants as compared to *lrpprc^+/+* (*Figure 4B*). To interpret if the decreased expression of mitochondrial transcripts was due to a decrease in mitochondrial number, we did mtDNA copy number analysis on *lrpprc* homozygous mutants. mtDNA copy number was not observed to be significantly decreased in *lrpprc^GBT0235/GBT0235* mutants as compared to wild-type (*Figure 4—figure supplement 1*). These findings are consistent with data observed in the previous studies that have shown a crucial role for *lrpprc* in the mitochondrial mRNA stability (*Gohil et al., 2010*). Homozygous mutants were also interrogated to assess the levels of lactate, a clinical mitochondrial marker that is observed to be altered in patients presenting with LSFC symptoms. Analysis of zebrafish lysate using gas chromatography/mass spectrometry (GC/MS) revealed increased levels of lactate in *lrpprc^GBT0235/GBT0235* mutants (*Figure 4C*).

Survivors of the early life metabolic crises associated with LSFC have multisystemic involvement and often show skeletal muscle phenotypes including hypotonia, muscle weakness, and mobility defects (*Sasarman et al., 2015*). To investigate whether our *lrpprc^GBT0235/GBT0235* mutants recapitulate this phenotype, we used birefringence as a readout of skeletal muscle development and structure. Four dpf *lrpprc^GBT0235/GBT0235* larvae exemplified (*Figure 5A and B*) a decrease in integrated density and mean gray value when compared with *lrpprc^+/+* siblings (*Figure 5C-E*) and indicated a deficiency in muscle development specific to *lrpprc^GBT0235/GBT0235* larvae. Assessment for the muscle phenotype in the 4 dpf liver-rescued homozygous mutants displayed a similar decrease in the parameters for muscle integrity such as integrated birefringence density as compared to homozygous mutants (*Figure 5—figure supplement 1*).

A subset of patients also displays neural phenotypes such as lesions in the brainstem and basal ganglia. To investigate for any evidence of neural necrosis, acridine orange (AO) staining was performed on *lrpprc* homozygous animals and wild-type siblings. Performing a blindfold analysis on the representative images across both genotypes, we didn't notice any significant difference in the necrotic signal represented by acridine orange spots (*Figure 5—figure supplement 2*).

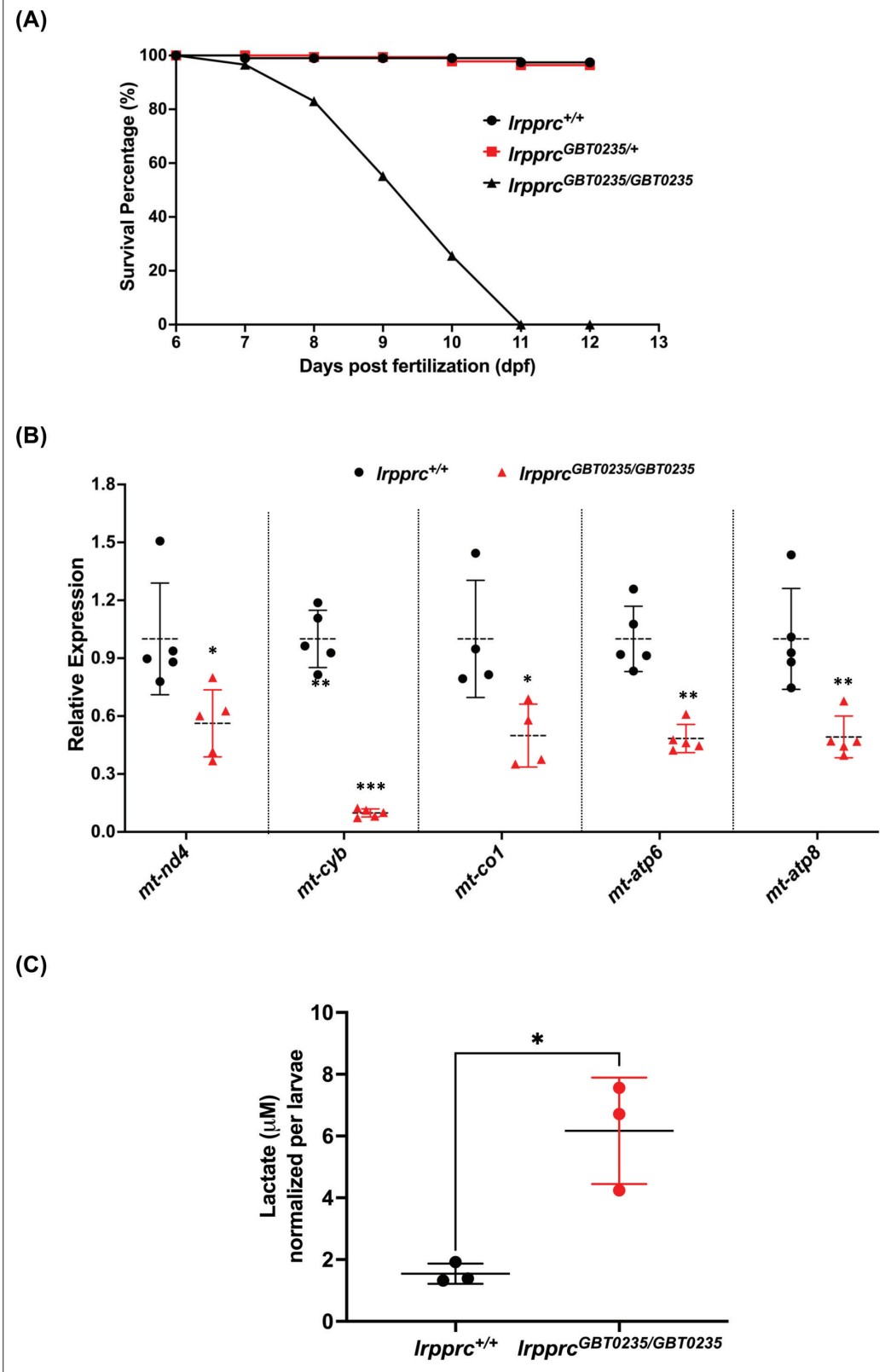

**Figure 4.** *lrpprc^{GBT0235/GBT0235}* mutants recapitulate the hallmarks of LSFC. (**A**) Survival percentage of *lrpprc^{+/+}*, *lrpprc^{GBT0235/+}*, and *lrpprc^{GBT0235/GBT0235}*. Data is represented from independent experiments (N=5 pairs) (***Figure 4— source data 1***) (**B**) Relative expression of mtDNA transcripts in the *lrpprc^{GBT0235/GBT0235}* and *lrpprc^{+/+}* assessed by qRT-PCR. Mitochondrial transcripts were normalized to *eef1a1l1* transcript levels. The black circle represents wild-

*Figure 4 continued on next page*

*Figure 4 continued*

type and the red triangle represents homozygous mutants. Each data point represents embryos from the different clutch. Error bars are represented as SD (*: p-value < 0.05, **: p-value < 0.01, ***: p-value < 0.001). p-Values were determined by unpaired t-test. p-values (*mt-nd4*=0.0248; *mt-cyb*=0.0001; *mt-co1*=0.0371; *mt-atp6*=0.0011; *mt-atp8*=0.0089). (*Figure 4—source data 1*) (**C**) Relative lactate levels in the whole-body lysates of wild-type and *lrpprc* homozygous siblings. Lactate levels were normalized to the number of larvae. Error bars are represented as SD (*: p-value = 0.0392). p-Value was determined using the unpaired t-test. (*Figure 4—source data 1*).

The online version of this article includes the following source data and figure supplement(s) for figure 4:

**Source data 1.** Numeric data for the survival percentage, relative expression of mtDNA transcripts and relative lactate levels in *lrpprc* homozygous mutants.

**Figure supplement 1.** *Relative mtDNA copy number in* lrpprc *homozygous mutants:* No significant change was observed in the homozygous mutants as compared to wild-type siblings (*lrpprc*[+/+] *vs lrpprc*[GBT0235/GBT0235]; p-value = 0.8263).

## *lrpprc*[GBT0235/GBT0235] mutants display altered transcriptomic signature

To assess genome-wide transcriptional changes in *lrpprc*[GBT0235/GBT0235] mutants, strand-specific paired-end RNA sequencing was performed (Illumina, USA) with a mean read length of 150 bp, and ~70–183 million reads were generated for both the *lrpprc*[+/+] and *lrpprc*[GBT0235/GBT0235] mutants. The reads were aligned onto the Zv10 reference genome after filtering low-quality reads with cut-off Q30. Further, the reads were pseudoaligned with an average mapping percentage of ~81%. Empirical cutoff of genes with $\log_2$Fold-Change $\geq 1$ and $\log_2$Fold-Change<=-1 was shortlisted (p-value$_{adj}$<0.05) as upregulated and downregulated genes respectively (*Figure 6A*). 88 genes were observed to be significantly upregulated, and 176 genes were significantly downregulated in *lrpprc*[GBT0235/GBT0235]. Overall, 12 and 16 human orthologs of zebrafish upregulated and downregulated genes, respectively overlapped with the human Mitocarta 3.0 database (*Rath et al., 2021*; *Figure 6B*). Gene set enrichment analysis predicted oxidative phosphorylation and glycerolipid metabolism pathways to be depleted in the *lrpprc*[GBT0235/GBT0235] mutants (*Supplementary file 2*). These pathways followed a similar trend as observed by the expression of mitochondrial encoded transcriptome and *lrpprc* transcript.

Functional classification analysis of the differentially expressed genes using PANTHER (*Mi et al., 2019*) was performed on two bases: protein class based on the biochemical property and the other was the biological process (*Figure 6C*). Protein classes that were majorly represented were metabolite interconversion enzymes and protein modifying enzymes. Most of the differential expressed genes were involved in biological processes such as the cellular process, metabolic process, and biological regulation.

## *lrpprc*[GBT0235/GBT0235] mutants show altered lipid metabolism

The liver is a tissue that has high energy demand and thus mitochondria are abundant in hepatocytes. The spectrum of the expression of the Lrpprc-mRFP protein was well captured in the liver of the *lrpprc*[GBT0235/+] and *lrpprc*[GBT0235/GBT0235] mutants (*Figure 7A*). Interestingly, *lrpprc*[GBT0235/GBT0235] mutants further display a darkened liver phenotype under brightfield microscopy, reinforcing the involvement of the liver as a distinguishing feature in LSFC as compared to traditional Leigh Syndrome (*Figure 7B*). Phenotype such as higher contrast liver was used as the basis to investigate the role of the liver in the progression or management of this disorder. Oil Red O staining revealed an abnormal distribution of lipids with increased lipid accumulation in the livers of *lrpprc*[GBT0235/GBT0235] mutants (*Figure 7C and D*) compared to *lrpprc*[+/+], suggesting the observed darkened liver phenotype could be caused by a change in the density of the stored lipids. To ascertain the hepatic mitochondrial stress which leads to larval lethality (*Figure 7E*), electron microscopy of the hepatocytes was conducted for homozygous mutants and their wild-type siblings to assess the mitochondrial structure. Altered morphology of the mitochondria was observed in the hepatocytes of the *lrpprc* homozygous mutants (*Figure 7F*).

Since lipid metabolism is a basic liver function dependent on mitochondrial activity, and disruption may lead to acute metabolic crises and death, we further examined how lipid profiles are altered in the null mutants using earlier described methods (*Quinlivan et al., 2017*). By coupling the feeding of a fluorescent fatty acid analog with high-performance liquid chromatography methods, we asked whether the *lrpprc* null mutants have differences in fatty acid metabolism compared to wild-type. For these experiments, 6 dpf larvae were fed with BODIPY FL C12 (lipid isotopic label) and TopFluor

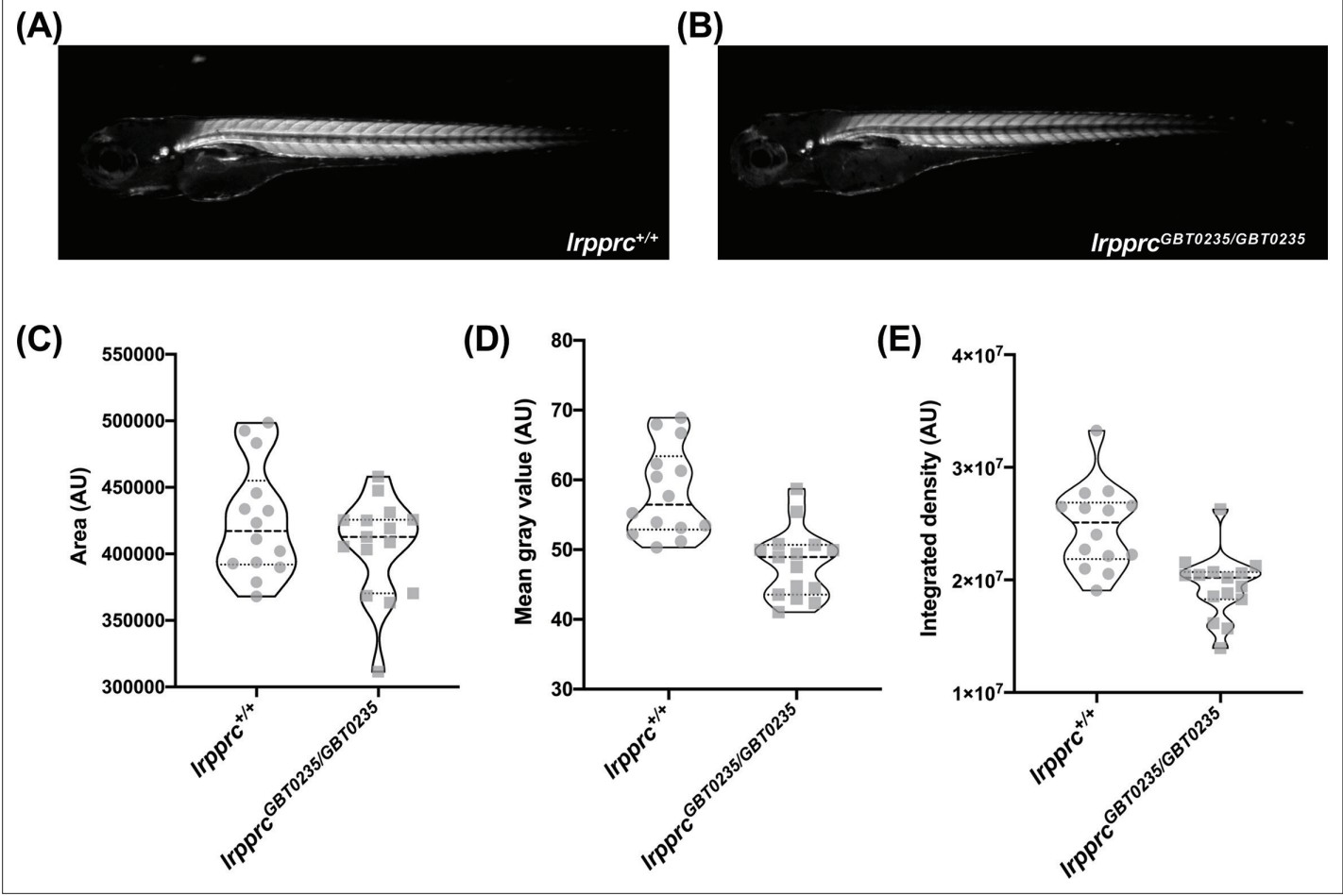

**Figure 5.** *lrpprc*^GBT0235/GBT0235 mutants display decreased birefringence. (**A–B**) Representative Birefringence images of *lrpprc*^+/+ (**A**) and *lrpprc*^GBT0235/GBT0235 mutants (**B**). (**C–E**) The images and graphs in the figure show the birefringence area of the region of interest (ROI) (**C**), mean gray value (**D**), and integrated density (**E**) between *lrpprc*^+/+ and *lrpprc*^GBT0235/GBT0235 mutants. *lrpprc*^GBT0235/GBT0235 mutants display similar birefringence area (p-value = 0.4773) but a decrease in mean gray value (p-value < 0.0001) and integrated density (p-value = 0.0001). Each individual data point represents a single embryo (For *lrpprc*^+/+; N=14 and *lrpprc*^GBT0235/GBT0235; N=15). Each parental pair represents a biological replicate. p-Values were determined using the Mann-Whitney test. (***Figure 5—source data 1***) (magnification- 5×).

The online version of this article includes the following source data and figure supplement(s) for figure 5:

**Source data 1.** (**A**) Numeric data for the birefringence measurements in *lrpprc*^+/+ and homozygous mutant larvae, *lrpprc*^GBT0235/GBT0235; (**B**) numeric data for the birefringence measurements in 4 dpf *lrpprc*^+/+, homozygous mutant larvae, *lrpprc*^GBT0235/GBT0235, and liver-specific rescued larvae, *Tg(fabp10:Cre)lrpprc*^GBT0235/GBT0235 ; (**C**) number of acridine orange particle counts across *lrpprc* homozygous mutants and wild-type siblings.

**Figure supplement 1.** *lrpprc*^GBT0235/GBT0235 and *Tg(fabp10:Cre)lrpprc*^GBT0235/GBT0235 mutants display decreased birefringence at 4 dpf.

**Figure supplement 2.** *lrpprc* homozygous mutants do not display neuronal necrosis.

Cholesterol (a fluorescent lipid that cannot be esterified) to correct for differences in the amounts ingested by individual larvae. Total lipid extracts were then subjected to HPLC with fluorescent detection.

Chromatograph peak areas were observed to be lower in the homozygotes indicating less total lipids were ingested, and the non-metabolizable fluorescent cholesterol analog allowed us to correct for these differences (***Figure 8A and B***). After chromatograph peak areas were normalized to the non-metabolizable lipid, we found that the *lrpprc*^+/+, *lrpprc*^GBT0235/+, and *lrpprc*^GBT0235/GBT0235 larvae channeled the dietary fluorescent FA into cholesterol ester in similar levels, allowing for efficient cholesterol transport. However, we found that the null mutants incorporated twice as much of the dietary fluorescent FA into non-polar lipids compared to their wild-type siblings [*lrpprc*^GBT0235/GBT0235/*lrpprc*^+/+=2.040; 95% CI = 1.122–3.709; p-value = 0.019; (***Figure 8C***)].

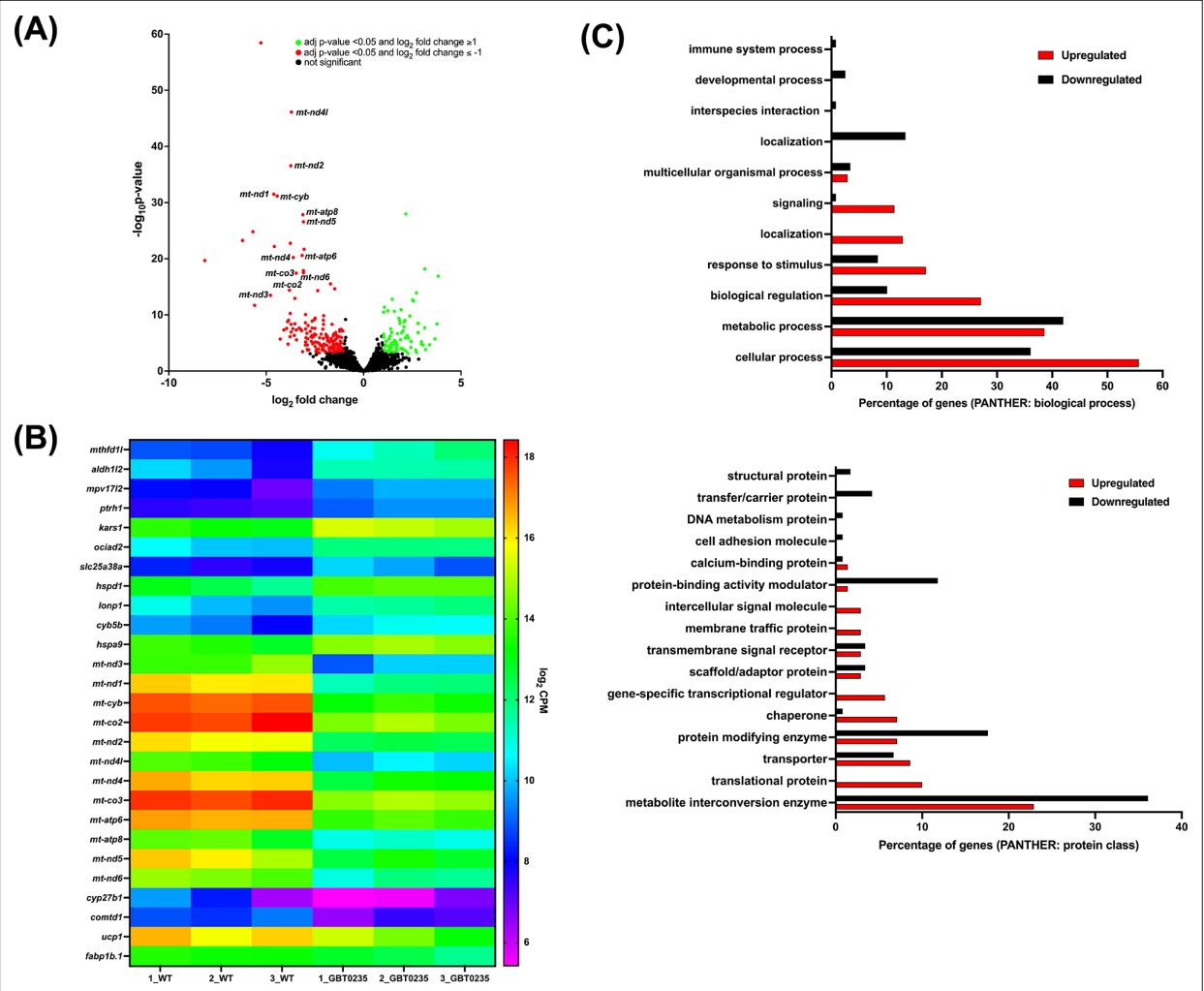

**Figure 6.** RNAseq of *lrpprc*<sup>GBT0235/GBT0235</sup> homozygous mutants. (**A**) Volcano plot of differentially expressed genes in the homozygous mutants. $\log_2$ of fold change and minus of $\log_{10}$ of p-value is represented on the x-axis and y-axis, respectively. The red dots signify the significantly downregulated genes and the green dots represent the significantly upregulated genes between the 6 dpf *lrpprc*<sup>GBT0235/GBT0235</sup> and *lrpprc*<sup>+/+</sup> larvae. Non-significant genes are represented by black dots (**B**) Heat map visualization of expression of zebrafish orthologs for human Mitocarta genes. The gradient color scale represents the $\log_2$CPM value obtained for each of the zebrafish mitochondrial orthologs across the biological replicates for each genotype. (**C**) PANTHER classification for all the significantly differentially expressed genes in the homozygous mutant according to protein class and biological process. Each histogram represents the percentage of genes having hits in the PANTHER database that fall in each of the categories, that is, biological process, and protein class.

We then looked for the contribution of any particular peak or peak cluster towards higher levels of incorporation into non-polar lipids in the homozygous mutants and found that three of four non-polar lipid peaks or peak clusters are significantly higher *lrpprc*<sup>GBT0235/GBT0235</sup>/*lrpprc*<sup>+/+</sup> for peak 1, peak 2 and peak 4, respectively = 1.73 [1.02–2.94] (p-value = 0.043), 1.99 [1.13–3.52] (p-value = 0.017), and 2.91 [1.47–5.79] (p-value = 0.002) (***Figure 8B***). While we are unable to define the exact identities of these peaks at this time, these are likely to be largely made of triglycerides and diglycerides, the major components of nonpolar lipids along with cholesterol esters. The fatty acid (FA) composition, as detected by HPLC-CAD, did not differ between the groups (post-normalization for total lipid levels).

## Liver-specific rescue reverses phenotypes seen in *lrpprc*<sup>GBT0235/GBT0235</sup> mutants

The hepatic dysfunction seen in LSFC patients coupled with the lipid accumulation and darkened liver phenotype in GBT0235 mutants pointed to the liver as an important potential therapeutic avenue.

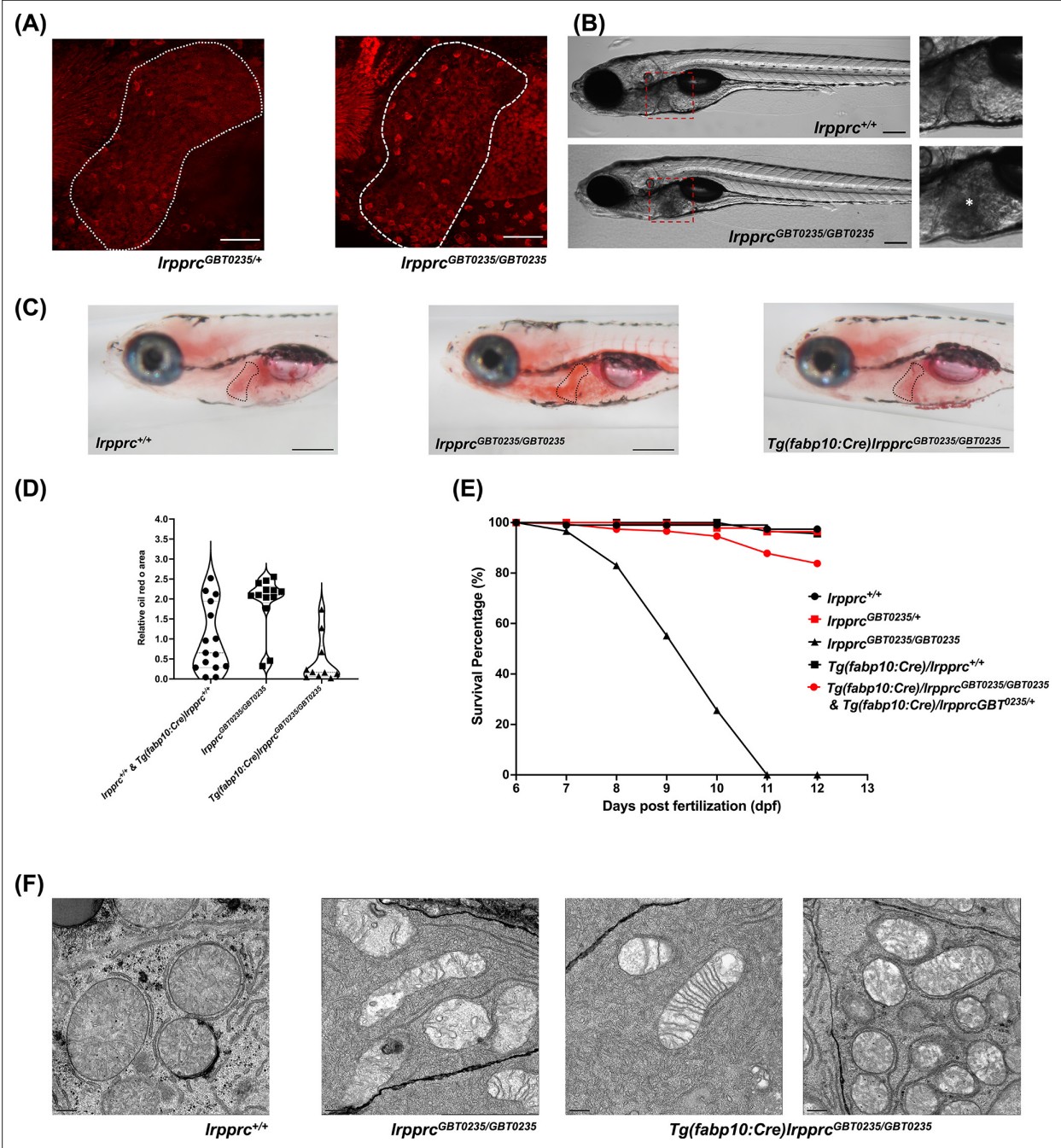

**Figure 7.** The liver plays an important role in the pathology of LSFC and genetic liver-specific rescue rescues the lipid defect and mortality in *lrpprc* homozygous mutant larvae. (**A**) Representative images of the livers of 6 dpf old *lrpprc*GBT0235/+ and *lrpprc*GBT0235/GBT0235 mutants at 40× (scale bar: 50 μm). (**B**) Brightfield image of 6 dpf *lrpprc*+/+ and *lrpprc*GBT0235/GBT0235 mutants. Homozygous mutants display a dark liver phenotype as compared to the wild-type controls. The region showing dark liver has been marked by an asterisk (scale bar: 150 μm). (**C**) Oil red O staining for assessment of lipid accumulation in the 6 dpf mutants and rescued larvae. Increased lipid accumulation was observed in the homozygous mutants compared to wild-type larvae. In the liver-specific rescued homozygous *lrpprc* mutants, *Tg(fabp10:Cre)lrpprc*GBT0235/GBT0235, no accumulation of lipids was observed (scale bar: 200 μm). (**D**) The graph show increase in the relative area of the oil red stain for the accumulated lipids, between wild-types (non-rescued wild-type, *lrpprc*+/+ and liver-specific rescued wild-type *Tg(fabp10:Cre)lrpprc*+/+) and *lrpprc*GBT0235/GBT0235 mutants, indicating an increase in the lipid content (p-value = 0.0083). The levels are restored in homozygous rescued larvae, *Tg(fabp10:Cre)lrpprc*GBT0235/GBT0235 (p-value < 0.0001). There was no significant difference between wild-types (non-rescued wild-type, *lrpprc*+/+ and liver-specific rescued wild-type *Tg(fabp10:Cre)lrpprc*+/+) and liver-specific rescued mutants, *Tg(fabp10:Cre) lrpprc*GBT0235/GBT0235 (p-value = 0.0623). p-Values were determined by the Mann-Whitney test. Each data point represents a single embryo. For, *lrpprc*+/+ N = 8; *Tg(fabp10:Cre)lrpprc*+/+ N = 7; *lrpprc*GBT0235/GBT0235 N=13; *Tg(fabp10:Cre)lrpprc*GBT0235/GBT0235, N=10. (**Figure 7—source data 1**) (**E**) Liver-specific rescued mutants display an improved survival rate beyond 11 dpf (**Figure 7—source data 1**). (**F**) Representative electron micrographs of the mitochondria in

*Figure 7 continued on next page*

*Figure 7 continued*

hepatocytes for 8 dpf *lrpprc* wild type, *lrpprc* homozygous mutants, and liver-specific *lrpprc* rescued larvae. Altered mitochondrial morphology displayed by *lrpprc*[GBT0235/GBT0235] was observed to be improved in the rescued mutants, *Tg(fabp10:Cre)lrpprc*[GBT0235/GBT0235] (scale bar: 0.5 µm).

The online version of this article includes the following source data and figure supplement(s) for figure 7:

**Source data 1.** (**D**) Numeric data for the oil red area in *lrpprc*[+/+], homozygous mutant larvae, *lrpprc*[GBT0235/GBT0235,] and liver-specific rescued homozygous mutant larvae, *Tg(fabp10:Cre)lrpprc*[GBT0235/GBT0235]; (**E**) data for the survival percentage of *lrpprc*[+/+], *lrpprc*[GBT0235/GBT0235] and larvae, *Tg(fabp10:Cre)lrpprc*[GBT0235/GBT0235].

**Figure supplement 1.** Irreversible liver-specific Cre recombinase-mediated rescue.

The reversible nature of the GBT construct enables the organismal or tissue-specific reversion to wild type by removal of the cassette via *Cre* recombinase. Therefore, by crossing *lrpprc* mutants to *Tg(−2.8fabp10:Cre;−0.8cryaa:Venus)*[S955] zebrafish, we were able to create a liver-rescued *lrpprc* zebrafish, *Tg(fabp10:Cre)/lrpprc*[GBT0235/GBT0235] line that could then be scored for the loss of liver-specific mRFP (**Figure 7—figure supplement 1**) and tested for reversion of the homozygous mutant phenotypes. Expression of *fabp10* has been previously shown to be restricted to the liver in the developing zebrafish larvae (**Bernard Thisse, 2004**; **Wu et al., 2017**). Following the generation of this line, we replicated the survival experiments to assess lipid metabolism to determine whether we would see an improvement in liver function. Overall, we noted a decrease in the cholesterol and neutral lipid accumulation in the Oil Red O staining in the liver rescue conditions (**Figure 7D**). Moreover, we saw an improved survival rate of the rescued mutants surviving until at least 12 dpf (**Figure 7E**). Liver-specific rescue in the homozygous mutant background mostly reverted the mitochondrial morphology to an extent (**Figure 7F**).

## The altered dietary lipid metabolism in *lrpprc*[GBT0235/GBT0235] mutants is restored to wild-type levels through liver-specific rescue methods

Since it was observed that lipid metabolism was altered in the *lrpprc* null mutants using HPLC methods, we checked whether the liver-specific rescue restores metabolic function to wild-type levels. Not only did the liver-specific rescued null mutants ingest as much as their wildtype and heterozygous siblings, as measured by their non-metabolizable TopFluor Cholesterol peak area, but the total peak area of nonpolar lipids was also lowered to that of the wildtype siblings [*lrpprc*[GBT0235/GBT0235]; *Tg(fabp10:Cre)/lrpprc*[GBT0235/GBT0235]=0.542; 95% CI = 0.357-.823; p-value = 0.004; (**Figure A, 8B and 8C**)]. The genetic background of the liver-specific CRE rescue did not contribute to any change in lipid levels (*Tg(fabp10:Cre)lrpprc*[+/+]:*lrpprc*[+/+]=0.944; 95% CI = 0.640–1.391; p-value = 0.769 **Figure 8A and D**). p-Values were obtained from the standard normal z-table.

## Discussion

In this study, we present a novel zebrafish model for Leigh Syndrome, French Canadian Type (LSFC) identified through the RP2 gene-breaking transposon insertional mutagenesis screen. Phenotypic assessment of this novel allele demonstrates how zebrafish serve as an excellent model to study LSFC. Zebrafish Lrpprc shares 51% homology with its human ortholog and has 31 predicted similar PPR domains indicating the conserved mitochondrial mRNA binding role of this gene. The spatiotemporal analysis of Lrpprc-mRFP shows a co-localization with an established mitochondrial marker in the caudal fin region and myocytes, as the truncated fusion proteins retain the mitochondrial targeting sequence maintaining its predicted subcellular localization. We also observed remarkable ubiquitous expression patterns throughout the organism and hotspots in the liver, skeletal muscle, and eye. In addition, increased expression of the fusion protein in certain tissues such as the liver, skeletal muscle, and eyes is indicative of the observation that mitochondrial density is tied to energy-intensive processes like metabolism, movement, and vision. Differential expression data will nevertheless help in expanding our understanding of the different roles of this gene in tissue-specific niches.

Sequence analysis of the predicted zebrafish Lrpprc protein revealed the presence of some known homology domains like secretory (SEC)1, RPN7, DUF2517, and EAD11. SEC1 homology domain-containing proteins are known to be involved in vesicular trafficking, synaptic transmission,

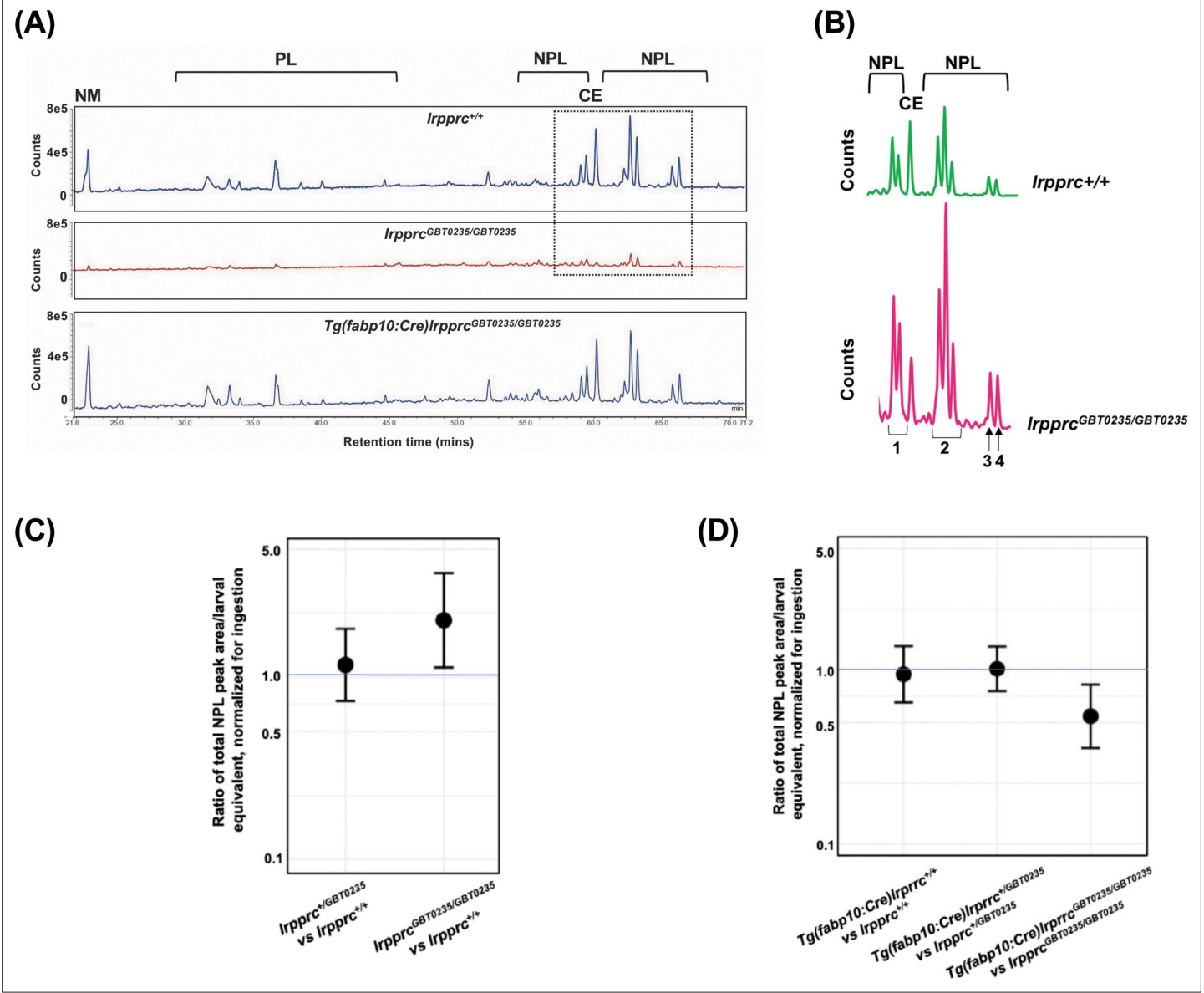

**Figure 8.** Genetic liver-specific rescue of altered dietary lipid metabolism in *lrpprc* homozygous mutant larvae. (**A**) Representative chromatographs are shown: lipid extractions from whole wild-type larvae (top) and whole *lrpprc* homozygous mutant larvae (middle). Peaks represent different lipid species and are quantified by peak area. Liver-specific rescue of *lrpprc*$^{GBT0235/GBT0235}$ mutants restores nonpolar lipid levels to wild-type levels in liver-specific rescued homozygous mutant larvae (bottom). All peaks were normalized to the amount of non-metabolizable fluorescent reagent ingested (NM). (**B**) Excerpts from representative chromatographs, wild-type larvae (top), and *lrpprc* homozygous mutant larvae (bottom). Peaks or peak areas (labeled as 1–4) were individually analyzed for contribution to the overall higher NPL level in *lrpprc* homozygous mutants compared to their wild-type siblings. (**C**) 95% CI plot: *lrpprc*$^{GBT0235/GBT0235}$ generated 2.04 times more non-polar lipids compared to their wild-type siblings (*lrpprc*$^{GBT0235/GBT0235}$/*lrpprc*$^{+/+}$=2.040, p-value = 0.019). p-Value was obtained from the standard normal z-table. (**D**) 95% CI plot: *Tg(fabp10:Cre)lrpprc*$^{GBT0235/GBT0235}$ restored the levels of non polar lipids as compared to *lrpprc*$^{GBT0235/GBT0235}$ homozygous mutants. PL = phospholipids; NPL = nonpolar lipids (triglycerides, diglycerides); CE = cholesteryl ester; NM = normalizer for amount eaten. (***Figure 8—source data 1***).

The online version of this article includes the following source data for figure 8:

**Source code 1.** Code used to perform generalized estimated equations (gee) comparing total non polar lipid area as well as individual peaks/groups of peaks.

**Source data 1.** Data for the peak areas per larval equivalent and normalized to the TopFluor cholesterol peak across *lrpprc*$^{+/+}$, *lrpprc*$^{GBT0235/GBT0235}$, and larvae, *Tg(fabp10:Cre)lrpprc*$^{GBT0235/GBT0235}$.

and general secretion (*Liu and McKeehan, 2002*; *Halachmi and Lev, 1996*). In addition, the human LRPPRC protein interacted with various other protein partners like fibronectin through its SEC1 domain (*Liu and McKeehan, 2002*). The effector-associated domains (EADs) are small, non-catalytic domains and are predicted to be involved in protein-protein interactions (*Kaur et al., 2021*). The RPN7 domain is one of the regulatory subunits of the 26 S proteasome and is known to be required for structural integrity in *Saccharomyces cerevisiae* (*Isono et al., 2004*). The region of zebrafish Lrpprc from amino acids 966–1012 had homology towards the protein of unknown function (DUF2517) domain, which is common to proteobacteria. The function of the DUF2517 homology domain is still unknown. Our in silico structural studies demonstrated that the C-terminus of human LRPPRC was well aligned with the C-terminus of zebrafish Lrpprc (*Figure 2—figure supplement 3*), strengthening the hypothesis that the zebrafish Lrpprc is playing a critical role in cell metabolism as discussed above. Additionally, the multiple sequence alignment of LRPPRC showed that the C-terminus region of this protein was conserved between several model organisms (*Figure 2—figure supplement 3*). Primary sequence and structural analyses highlighted the loss of domains important for maintaining the stability of mtRNA transcripts such as RPN7, DUF2517, SEC1, and EAD11 present in the C-terminus region of LRPPRC where a protein truncation led to LSFC-associated phenotype in zebrafish.

To understand the phenotypic and metabolic findings in the Lrpprc-deficient zebrafish model, we also analyzed the potential RNA binding targets of the Lrpprc protein. Lrpprc contains multiple PPR motifs, and these functional domains determine the target RNA-binding sites. We first annotated the PPR motifs for the Lrpprc protein for zebrafish and human orthologs to explore RNA binding sites. Preliminary analysis using the TPRpred server identified 22 PPR motifs for both human and zebrafish Lrpprc proteins (*Hammani et al., 2016*). TPRpred is a profile-based method that uses a *P*-value-dependent score offset and exploits the tendency of repeats to occur in tandem (*Karpenahalli et al., 2007*). This tool not only predicts the Tetratrico Peptide Repeat (TPR) motifs, but also the related PPRs and SEL1-like repeats. However, a study by Spåhr and colleagues had predicted 33 PPR motifs for the human LRPPRC protein (*Spåhr et al., 2016*) which was more than what was predicted by the TPRpred (*Karpenahalli et al., 2007*) server. Additionally, we evaluated the secondary structure containing the Lrpprc proteins by PSIPRED server, we found that both humans and zebrafish are all α-helical and may contain more hairpin of α-helices than what TPRpred predicted. To address this discrepancy and to conduct a meticulous study to systematically identify PPR motifs in human and zebrafish Lrpprc, we employed a deep-learning-based structure modeling tool, RoseTTAFold (*Baek et al., 2021*). Using RoseTTAFold we designed a workflow to predict 31 and 34 PPR motifs for the zebrafish and human orthologs respectively of Lrpprc proteins.

We investigated the sequences of the identified PPR motifs in each of the orthologs and deciphered the RNA recognition residues, which are typically found at the 5th and 35th positions in each PPR motif. These two amino acids play a major role in specifying the target nucleotide and constitute a code for nucleotide recognition. The potential target recognition codes for human and zebrafish Lrpprc PPR motifs were predicted to exhibit a range of nucleotide targets (*Figure 2—figure supplements 1 and 2*). We used the FIMO search algorithm to predict the potential binding sites for human and zebrafish Lrpprc proteins. *In silico* RNA-binding analysis revealed that transcripts coding for proteins involved in the complex IV activity such as *mt-co1* and *mt-co2* were among the hits with a high confidence score. Apart from complex IV genes, transcripts belonging to complex I and complex V were also identified to be recognized by the PPR motifs of the Lrpprc protein. There was a consensus for the majority of the target transcripts for the zebrafish and human orthologs indicating a conserved function of Lrpprc in the maintenance of the mitochondrial genome both in humans and zebrafish. Previously, human LRPPRC was reported to regulate transcripts coding for complex IV of mitochondrial electron transport chain (*Xu et al., 2004*; *Barchiesi and Vascotto, 2019*) which was very well aligned with these findings and supports the hypothesis that both human and zebrafish LRPPRC likely have similar mitochondrial targets. To the best of our knowledge, this is the first such analysis that has been performed and validated for PPR-based proteins.

Previously published reports have highlighted the role of PPR-containing proteins in mediating mRNA stability and translational activation (*Prikryl et al., 2011*; *Mesitov et al., 2019*). PPR

proteins have been shown to stabilize the RNA structure by blocking the access to exonucleolytic cleavage upstream and downstream of their binding sites in a way acting as protein caps at the 5' and 3' ends of the processed transcripts (*Prikryl et al., 2011*). It also confers a site-specific remodeling activity by sequestering the target RNA by preventing it from binding to the translation initiation region (*Prikryl et al., 2011*). Another functionality contributed by this class of proteins is aiding in the inhibition of uridylation of A-tailed transcripts, thereby preventing the degradation of mRNAs (*Mesitov et al., 2019*). These roles highlight the role(s) of PPR motif-containing proteins in the stability of the mitochondrially encoded transcriptome.

Upon further investigation using independent transcriptomics sequencing and targeted mitochondrial RNA qPCR, it was observed that downregulated mtDNA genes in the *lrpprc* homozygous zebrafish mutants overlapped with those predicted by the above algorithm. These results possibly explain the reduced expression of mitochondrial RNAs in the homozygous mutants, underscoring the role of Lrpprc as loss of this protein impedes the binding of this protein, therefore leading to mRNA instability and degradation.

As the pathophysiology of LSFC is not well understood, we set out to explore the physiological consequences of *lrpprc* loss-of-function mutation *in vivo* and whether we could mimic the clinical hallmarks seen in LSFC patients. Similar to human patients, *lrpprc*$^{GBT0235/GBT0235}$ mutants displayed complete larval lethality by 12 dpf. However, factors triggering the crisis and associated death in both patients and animal model was initially unclear. Consistent with previous *in vitro* studies (*Gohil et al., 2010*), expression of mitochondrial protein-coding transcripts was observed to be downregulated in these homozygous mutants and was further investigated using RNASeq, indicating LRPPRC to be a critical post-transcriptional regulator of mitochondrial transcripts. LRPPRC protein in mitochondria forms a complex with SRA stem-loop interacting RNA binding protein (SLIRP) protein in the mitochondrial matrix. The absence of LRPPRC leads to oligoadenylation of the transcripts. LRPPRC-SLIRP is also observed to relax RNA structure, making the 5' end of the mRNA accessible for the mitochondrial ribosome to initiate the translation (*Cui et al., 2019*). The mechanistic insights in our zebrafish *in vivo* model from the structure of Lrpprc protein and target sites of the predicted PPR domains as described earlier correlate with observations that have been well established in patient-derived cell culture model displaying the altered signature of mitochondrial transcripts (*Gohil et al., 2010*; *Sasarman et al., 2015*).

Decreased expression of these protein-coding transcripts is crucial for eliciting defects in oxidative phosphorylation and mitochondrial respiratory chain. These results were in concordance with the pathways responsible for mitochondrial biogenesis and homeostasis predicted to be depleted in the *lrpprc* homozygous mutants. Various functions including energy production, genome maintenance, ion/metabolite homeostasis, membrane dynamics, and transport of biomolecules can be attributed to approximately 1136 proteins residing in the mitochondria (*Rath et al., 2021*). We did note an overlap between the set of differentially expressed genes and those in MitoCarta catalogue, suggesting an altered mitochondrial transcriptome signature. Apart from classical respiratory chain subunits, we identified some key genes from this overlapped data to be differentially expressed that were responsible for pathways such as aldehyde metabolism, fatty acid metabolism, and most important, unfolded protein response. Genes belonging to fatty acid metabolism and cytochrome P450 were observed to be downregulated in the null mutants. *lrpprc* homozygous mutants exhibited an upregulation of chaperone proteins, heat shock protein 5 (*hspa5*), and heat shock 60_1 (*hspd1*), possibly eliciting mitochondrial unfolded protein response that forms an integral part of mitochondrial protein homeostasis.

Another high energy-requisition tissue on the phenotypic spectrum of LSFC that we explored was muscle. LSFC patients display muscle defects such as hypotonia and COX deficiency (*Oláhová et al., 2015*) and therefore, we adopted a birefringence measurement to assess the muscle phenotype in our null mutants. It takes a complex and highly organized structure (like the sarcomeres in skeletal muscle) to rotate the plane-polarized light and enable the visualization of birefringence. The noted decrease in birefringence in *lrpprc* mutants is most likely due to the disorganization of sarcomeres and other structures in skeletal muscle. It has also been noted that LRPPRC deficiency in patient-derived muscle cells resulted in combined complex I and complex IV deficiency (*Sasarman et al., 2015*). These skeletal muscle defects due to impaired *lrpprc* function may reflect the tissue-specific adaptation to mitochondrial bioenergetics response. The structural analysis

of identifying PPR motifs in zebrafish Lrpprc does explain some of these observations as it was revealed that the top potential binding sites of this protein were genes related to complex IV. A few of them were *mt-co1*, *mt-co2,* and *mt-co3* which encode subunits of the complex IV of the mitochondrial electron transport chain. Interestingly, muscle phenotypes that were displayed by *lrpprc* homozygous mutants were not mitigated by the liver-specific reversion of the mutagenicity cassette. This not only underscores the specificity of the revertible cassette in a given tissue but also indicates the improved survival and lipid homeostasis observed in rescued animals being at least partially independent of muscle dysfunction. One of the possible explanations for the impaired muscle function not contributing to lethality in rescued animals can be compensatory mitochondrial biogenesis or mitochondrial proliferation in muscle, a tissue with high metabolic demands that are also observed in patients with LSFC (*Cuillerier et al., 2021*). There might be adaptive OXPHOS compensation in these kinds of tissues to adapt to the physiological demands in models with a mutation in genes affecting global mtDNA translation leading to multi-complex deficiency. Further studies will be required to explain any of these possibilities exploring the role of bioenergetic adaptation happening in a specific tissue through unknown metabolite and protein players in the cellular milieu.

The liver-specific dysfunction coupled with the episodes of acute metabolic crises distinguishes LSFC from classic Leigh Syndrome (LS). A previous study suggested that liver dysfunction may factor into the increased lactic acidosis and metabolic disturbances as the liver play a key role in lactate homeostasis via the Cori cycle (*Sun et al., 2017*). In addition, it is important to note that other basic liver functions that are dependent on mitochondrial activity such as lipid metabolism, serum detoxification, and gluconeogenesis may be key to the development of metabolic crises since they have secondary effects in other organ systems. Mitochondrial morphology is the classical hallmark of underlying mitochondrial dysfunction. Mitochondria can undergo changes in shape accompanied by fusion and fission events which may involve remodeling at cristae junctions that dynamically appose and separate from each other (*Cogliati et al., 2016*; *Kondadi et al., 2020*). Under TEM, hepatic mitochondria had an altered morphology in the homozygous mutants, potentially impairing mitochondrial function. From the transcript studies and motif-based target prediction, we did observe a decreased expression of *mt-atp6* and *mt-atp8* transcripts due to Lrpprc deficiency. Proteins coded by these mRNAs constitute the complex V of the electron transport chain. It has been reported that complex V plays an essential role in defining the cristae morphology specifically in the areas of high membrane curvature at cristae ridges (*Cuillerier et al., 2017*).

It has been observed that mitochondrial disorders increase predisposition to intracellular lipid accumulation (*Morino et al., 2006*). Homozygous *lrpprc* mutants manifested early-onset phenotypes with hepatomegaly and progressive accumulation of dark-colored granules in the liver at 6 dpf. These granules appeared to be lipid droplets, which were validated with both Oil Red O staining and extensive *in vivo* feed/chase (i.e., metabolize) labeling lipid studies. Dietary lipid metabolism studies in the homozygous mutants revealed an accumulation of non-polar lipids such as triglycerides and diglycerides. Accumulation of these lipid species highlights an impaired lipid metabolism in the mutants similar to what is observed in LSFC patients (*Ruiz et al., 2019*). Metabolic diseases such as nonalcoholic fatty liver disease are associated with the accumulation of triglyceride levels and high low-density lipoprotein levels with an underlying mitochondrial dysfunction (*Simões et al., 2018*). Hepatic accumulation of such lipids may induce mitochondria to undergo dynamic changes such as depolarization and loss of ATP content (*Domínguez-Pérez et al., 2019*). Further probing of the transcriptomics study revealed that pathways for fatty acid degradation were depleted in these null mutants. The gene encoding for hepatic lipase_a (*Lipca*), a protein that plays an important role in triglyceride hydrolysis was downregulated by fold change of $\log_2$ ~2. Accumulation of triglyceride levels in homozygous mutants was corroborated with findings from the RNA-Seq study, where genes such as Cytochrome P450 Family 7 Subfamily A or Cholesterol 7 alpha-hydroxylase (*cyp7a1*) and *lipca* were observed to be downregulated. These genes are involved in the Farnesoid X receptor pathway that plays an important role in hepatic triglyceride metabolism (*Jiao et al., 2015*). Interestingly, we observed the targets for the sterol regulatory element-binding proteins (SREBPs) to be differentially expressed. Prominent amongst those were ELOVL fatty acid elongase 2 (*elovl2*), fatty acid desaturase 2 (*fads2*), and *lipca. Elovl2* is essential for lipid homeostasis, wherein it regulates the levels of

poly-unsaturated fatty acids preventing liver steatosis (*Pauter et al., 2014*). The binding proteins are thought to manage intracellular fatty acids and serve to provide them to specific enzyme pathways in the cytoplasm. Similarly, reduced *lipca* would result in fewer cytoplasmic fatty acids consistent with the less cellular requirement of fatty acid-binding proteins (also observed to be downregulated in our data). Reduced availability of fatty acids would allow fewer substrates for ELOVL2 and FADS2. Another lipid metabolism-associated gene observed to be downregulated was fatty acid-binding protein 10 a (*fabp10a*) by fold change of $\log_2$ ~2.27. Intrigued to explore, if the abolishment of *lrpprc* had an inter-organelle effect, we analyzed the expression of genes that are involved in peroxisome-associated pathways. To summarize, the transcriptomic analysis did reveal that *lrpprc* homozygous mutants displayed altered expression of genes that modify and regulate fatty acids and were consistent with less fatty acid reflux and metabolism observed in *lrpprc* homozygous mutants. Surprisingly, MPV17 mitochondrial inner membrane protein-like 2 (*mpv17l2*), one of the regulator genes for the expression of antioxidant enzymes (*Iida et al., 2006*), was upregulated by fold change of $\log_2$ 1.74. Overall, our zebrafish LSFC model exhibited a dysfunctional mitochondrial signature both at the transcriptional and phenotypic level, which was explored to design a genetic model therapy for LSFC-associated phenotypes.

The key translational and clinical utility of this study is the reversion of the LSFC-associated phenotype and survival using liver-specific Cre recombinase in the homozygous *lrpprc* mutants, demonstrating a critical role of this organ in the pathogenesis of disease in this animal model. Because GBT-RP2 contains two *loxP* sites flanking the protein trap component, we could exploit this utility to rescue the organ-specific function of *lrpprc*$^{GBT0235/GBT0235}$. Interestingly, reversion of both lethality and biochemical phenotypes was observed in the rescued mutants, suggesting the liver plays a critical role in the pathology of LSFC. Although the rescue of mortality phenotype in the Cre *lrpprc* homozygous mutants was remarkable, it should be noted that longitudinal studies are needed to follow up on the survival of the rescued larvae. However, there will be a few points such as variables in zebrafish husbandry settings and the expression of the transgenic recombinase in the hepatocytes that will need to be considered while designing such experiments. Further experiments are required to delineate the juvenile or adult-onset mortality in the rescued animals due to the expression of truncated Lrpprc protein in the whole body minus liver or due to constitutive expression of Cre recombinase in the liver.

Remarkably, these liver-specific rescued larvae exhibited similar levels of non-polar lipids as compared to the wild-type larvae, post-high-fat meal underscoring the lipid homeostasis as a key therapeutic paradigm in the progression of this disease. Mitigation of abnormal mitochondrial structure was also observed in the hepatocytes of the rescued larvae. Although we did observe that rescue was not completely reversible in the liver-specific Cre homozygous mutants. There may be several factors that can contribute to this observation such as incomplete penetrance or rescue of the mutant allele. As cristae are dynamic structures of mitochondria whose morphology is altered under various bioenergetics and physiological conditions (*Cogliati et al., 2016*; *Kondadi et al., 2020*), reversion of mutagenicity cassette in the hepatocytes is not sufficient to correct cristae alterations that appeared to be aggravated in the *lrpprc* homozygous mutants. A possible explanation for the restoration in the morphology may be attributed to the restoration of levels of mitochondrial transcripts and thus restoring mitochondrial-mediated liver functions such as beta-oxidation and fatty acid elongation.

With advances in gene editing alongside developments in both viral and non-viral vector delivery, there is hope for tissue-specific gene therapy as a potential avenue to help mitochondrial disease patients. Our findings highlight the potential therapeutic applications of advances in gene therapy for patients of LSFC by rescuing *lrpprc* expression in the liver. We anticipate that these results, together with those from the conditional knockout experiments will enhance our understanding of the function of the *lrpprc* gene in hepatic lipid homeostasis and shine a light on new therapeutic options.

# Materials and methods

**Key resources table**

| Reagent type (species) or resource | Designation | Source or reference | Identifiers | Additional information |
|---|---|---|---|---|
| Recombinant DNA reagent | pGBT-RP2.1 | *Clark et al., 2011*; *Ichino et al., 2020* | Addgene: 31828; GenBank: HQ335170.1 | *Figure 1* |
| Genetic reagent (*Danio rerio*) | Tg(−2.8fabp10:Cre; −0.8cryaa:Venus)$^{S955}$ | *Ni et al., 2012* | Provided by Dr. Wenbiao Chen, Vanderbilt University | *Figure 7(C–F)* and 8(A–D), *Figure 7—figure supplement 1* |
| Genetic reagent (*Danio rerio*) | Tg(MLS: EGFP) | *Kim et al., 2008* | Kind gift provided by Dr. Sridhar Sivasubbu (CSIR-IGIB) provided to him by Dr. Seok-Yong Choi, Chonnam National University | *Figure 3C–D, Figure 3—figure supplement 1* |
| Software, algorithm | RoseTTAFold | *Baek et al., 2021*; https://github.com/Rosetta Commons/RoseTTAFold | https://github.com/ RosettaCommons/ RoseTTAFold | *Figure 2A–C* |
| Software, algorithm | PyMOL (The PyMOL Molecular Graphics System, Version 2.5.2 Schrödinger, LLC) | https://pymol.org/2/ | | *Figure 2A–C* |
| Software, algorithm | CLUSTAL Omega v.1.2.4 | *Sievers et al., 2011* http://www.clustal.org/omega/ | | *Figure 2—figure supplement 3* |
| Software, algorithm | Jalview v.2.11.1.7 | *Waterhouse et al., 2009* https://www.jalview.org/ | | *Figure 2—figure supplement 3* |
| Software, algorithm | Pfam HMM database | https://pfam.xfam.org/ | | *Figure 2—figure supplement 3* |
| Software, algorithm | TPRpred | *Karpenahalli et al., 2007* https://toolkit. tuebingen.mpg.de/tools/tprpred | | |
| Software, algorithm | FIMO motif scanning program in the MEME Suite | *Grant et al., 2011* https://meme-suite.org/meme/ tools/fimo | | *Supplementary file 1B* and *Supplementary file 1C* |
| Other | Human reference mitochondrial genome (NC_012920.1) | https://www.ncbi.nlm.nih. gov/nuccore/251831106 | | *Supplementary file 1B* |
| Other | Zebrafish reference mitochondrial genome (NC_002333.2) | https://www.ncbi.nlm.nih. gov/nuccore/NC_002333.2 | | *Supplementary file 1C* |
| Instrument | Zebrafish embryonic genotyper | *Lambert et al., 2018* https://www.wfluidx.com/ | | |
| Commercial assay or kit | MyTaq Red polymerase | Meridian Bioscience | Catalog no: BIO-21105 | |
| Chemical compound, drug | Agarose | VWR Life Science | Catalog no: 0710 | |
| Chemical compound, drug | Tween-20 | Bio-Rad | Catalog no: 170–6531 | |
| Chemical compound, drug | Sodium hydroxide anhydrous | MP Biomedicals | Catalog no: 153495 | |
| Chemical compound, drug | Tris base | Millipore-Sigma | Catalog no: 11814273001 | |
| Chemical compound, drug | Phenylthiocarbamide | Millipore-Sigma | Catalog no: P7629 | |
| Chemical compound, drug | Tricaine | Millipore-Sigma | Catalog no: A5040 | |

*Continued on next page*

*Continued*

| Reagent type (species) or resource | Designation | Source or reference | Identifiers | Additional information |
|---|---|---|---|---|
| Chemical compound, drug | Low-melting agarose | Fisher Scientific | Catalog no: BP1360 | |
| Recombinant DNA reagent | pT3TS-NLS-BFP | This paper | | |
| Commercial assay or kit | T3 mMessage mMachine transcription kit | Thermo Scientific | Catalog no: AM1348 | |
| Commercial assay or kit | NEB Monarch RNA cleanup kit | New England Biolabs | Catalog no: T2040L | |
| Chemical compound, drug | PstI (restriction enzyme) | New England Biolabs | Catalog no: R0140S | |
| Chemical compound, drug | N,O,-bis-(trimethylsilyl) trifluoroacetamide with 1% trimethylchlorosilane (BSTFA +TMCS) | Thermo Scientific | Catalog no: TS-38834 | |
| Chemical compound, drug | Ethyl acetate | EMD-Millipore | Catalog no: EX0245-1 | |
| Chemical compound, drug | Sodium Chloride | Fisher Scientific | Catalog no: S271 | |
| Chemical compound, drug | 6 N Hydrochloric acid | EMD-Millipore | Catalog no: HX0603-75 | |
| Chemical compound, drug | O-(2,3,4,5,6-Pentafluorobenzyl) hydroxylamine Hydrochloride | TCI America | Catalog no: P0822 | |
| Chemical compound, drug | TRIzol | Thermo Scientific | Catalog no: 15596026 | |
| Chemical compound, drug | Chloroform | Millipore-Sigma | Catalog no: C2432 | |
| Chemical compound, drug | Ethanol | Millipore-Sigma | Catalog no: E7023 | |
| Chemical compound, drug | Ethylenediaminetetraacetic acid disodium salt dihydrate (EDTA) | Millipore-Sigma | Catalog no: E5134 | |
| Chemical compound, drug | Potassium chloride | Millipore-Sigma | Catalog no: P5405 | |
| Chemical compound, drug | NP-40 permeating solution | Alfa Aesar | Catalog no: J60838 | |
| Chemical compound, drug | Proteinase K | Qiagen | Catalog no: 19133 | |
| Chemical compound, drug | Dnase I | Qiagen | Catalog no: 79254 | |
| Commercial assay or kit | RNeasy Mini Kit | Qiagen | Catalog no: 74104 | |
| Commercial assay or kit | SuperScript III Reverse Transcriptase | Thermo Scientific | Catalog no: 18080093 | |
| Commercial assay or kit | SensiFAST SYBR Lo-ROX kit | Meridian Bioscience | Catalog no: BIO-94005 | |
| Chemical compound, drug | Acridine orange | Millipore-Sigma | Catalog no: A8097 | |
| Software, algorithm | FIJI | https://imagej.net/software/fiji/ | | |
| Software, algorithm | GraphPad Prism 8 | https://www.graphpad.com/ | | |

*Continued on next page*

*Continued*

| Reagent type (species) or resource | Designation | Source or reference | Identifiers | Additional information |
|---|---|---|---|---|
| Commercial assay or kit | NEBNext Ultra RNA Library Prep Kit | New England BioLabs | Catalog no: E7530 | Library preparation performed at GENEWIZ, LLC (USA) |
| Software, algorithm | Trimmomatic | *Bolger et al., 2014* | | RNA sequencing analysis |
| Software, algorithm | Kallisto | *Bray et al., 2016* | | RNA sequencing analysis |
| Software, algorithm | Tximport | *Soneson et al., 2015* | | RNA sequencing analysis |
| Software, algorithm | DESeq2 | *Love et al., 2014* Differential gene expression analysis | | RNA sequencing analysis |
| Software, algorithm | Zebrafish Information Network (ZFIN) | https://zfin.org/ | | |
| Software, algorithm | Ensembl | http://useast.ensembl.org/Danio_rerio/Info/Index | | |
| Software, algorithm | PANTHER | *Mi et al., 2019* http://www.pantherdb.org/geneListAnalysis.do | | |
| Software, algorithm | GeneTrail | *Stöckel et al., 2016* https://genetrail2.bioinf.uni-sb.de/ | | |
| Chemical compound, drug | Oil red o stain | Millipore-Sigma | Catalog no: O1391 | |
| Chemical compound, drug | 4% Paraformaldehyde | Alfa Aesar | Catalog no: J61899 | |
| Chemical compound, drug | Phosphate buffer saline | Millipore-Sigma | Catalog no: | |
| Chemical compound, drug | BODIPY FL C12 | Thermo Scientific | Catalog no: D3822 | |
| Chemical compound, drug | TopFluor Cholesterol | Avanti Polar Lipids | Catalog no: 810255 | |
| Software, algorithm | R | https://www.r-project.org/ | | |

## Zebrafish handling and husbandry

All adult zebrafish and embryos were maintained according to the guidelines established by Mayo Clinic Institutional Animal Care and Use Committee (IACUC number: A34513-13-R16).

## Mutant generation

The GBT0235 allele was generated by injecting the pGBT-RP2.1 cassette as described to create the *in vivo* protein trap lines (*Ichino et al., 2020*; *Clark et al., 2011*). Generation of the transgenic mutant fish lines involved microinjection of the pGBT-RP2.1 vector DNA together with the transposase RNA (Tol2) in single-cell zebrafish embryos, followed by the screening of the injected animals for GFP and RFP expressions. To generate the larvae for liver-specific rescue data, non-GBT transgenic marker lines carrying liver-specific *Cre* recombinase *Tg(−2.8fabp10:Cre; −0.8cryaa:Venus)$^{S955}$* (*Ni et al., 2012*) were used. *Tg(MLS: EGFP)* transgenic line was used to generate larvae to study the subcellular localization of the truncated fusion protein (*Kim et al., 2008*).

## Structure-based prediction of RNA binding targets for Lrpprc protein

To predict the target transcripts of human and zebrafish Lrpprc, we first calculated the AI-based models for both human and zebrafish full-length Lrpprc proteins using RoseTTAFold. RoseTTAFold is a three-track neural network-based system and has enabled us to predict the structures of the proteins at a high level of accuracy and detail (*Baek et al., 2021*). Both the molecular models were computed by using the mforge servers of Mayo Clinic hosted by the University of Illinois Urbana-Champaign (UIUC). The resulting structure files were then rendered with PyMOL (The PyMOL Molecular Graphics

System, Version 2.5.2 Schrödinger, LLC). Multiple sequence alignment of different LRPPRC sequences from different model organisms was conducted using CLUSTAL Omega v.1.2.4 (*Sievers et al., 2011*) by taking the default parameters and visualized by using Jalview v.2.11.1.7 (*Waterhouse et al., 2009*) software. The homology domain organization was predicted by using the Pfam HMM database search (http://pfam.xfam.org/). PPR motifs were predicted and annotated using the TPRpred (*Karpenahalli et al., 2007*) server and also by independently assessing the zebrafish and human Lrpprc predicted structures.

We used this structural information and the PPR cipher code to identify consensus binding sequences (*Figure 2—figure supplement 1* and *Figure 2—figure supplement 2*). We determined the putative target nucleotide sequence by using the identities of the amino acids at the 5th and 35th positions of each PPR motif of human and zebrafish Lrpprc proteins. We used the IUPAC degenerate nucleotide codes to obtain a consensus nucleotide sequence. These consensus sequences were then used to identify potential binding targets of human and zebrafish Lrpprc by deploying the FIMO motif scanning program in the MEME Suite (https://meme-suite.org/meme/tools/fimo). The FIMO program searches for the occurrences of known motifs in the sequence databases. The putative nucleotide sequence was then used to search human and zebrafish Lrpprc RNA binding sites against the mitochondrial genomes of humans (NC_012920.1) and zebrafish (NC_002333.2), respectively using the FIMO program. The predicted target sites were ranked by p- values calculated by FIMO (*Grant et al., 2011*).

## Zebrafish embryo genotyping (ZEG) protocol

The deficiency of LRPPRC in humans shows an early neonatal phenotype, necessitating the ability to investigate GBT0235 mutants at the early stages of life. Likewise, our *lrpprc*$^{GBT0235/GBT0235}$ mutants exhibit lethality starting around 8 dpf. We needed the ability to genotype larvae at a young age without sacrificing to enable experiments to be efficiently conducted on living animals such as the survival assays and HPLC lipid analysis. An automated genotyping device (*Lambert et al., 2018*), non-invasive enzymatic genotyping. (*Sabharwal et al., 2021*; *Zhang et al., 2020*), and larval fin clip method (*Wilkinson et al., 2017*) were employed for the rapid cellular extraction of DNA from zebrafish embryos for various downstream experiments during the course of the study.

Larvae at 3 days post fertilization (dpf) were rinsed three times in fresh embryo water and transferred to a new petri dish. Larvae were aspirated in 12 µL of embryo media and placed on extraction chips in individual wells. An evaporation cover was attached, and the larvae were agitated using a vibrating motor that was powered with 2.4 V for 10 min. Immediately following the vibration period, the samples were removed from the chip and placed into strip tubes for storage. Larvae were replenished with embryo water and individually transferred to a 24-well plate for storage at 28 °C. The collected sample was used as a template for PCR amplification for genotyping of the larvae. Two PCR reactions were run using the same thermal cycler conditions. The first consisted of 5 µL of 5 x MyTaq Red PCR Buffer, 1 µL of 10 µM *lrpprc* ex 22 FP, 1 µL of 10 µM *GBT0235* RP, 0.25 µL of MyTaq DNA Polymerase, 3.5 µL of template, and 14.75 µL dH2O. The second PCR setup replaced the 10 µM *lrpprc* ex 22 FP RP with mRFP RP. Samples were run in a thermal cycler with following PCR conditions: (step 1) 95 °C, 5 min; (step 2) 95 °C, 1 min; (step 3) 61.3 °C, 30 sec; (step 4) 72 °C, 1 min; (step 5) Go to step 2 39 X; (step 6) 72 °C, 5 min; (step 7) 12 °C, hold. Samples were run on a 1% agarose gel and analyzed. We tested our locus-specific *lrpprc* primers (*Supplementary file 1A*) and found successful amplification of both our *lrpprc*$^{+/+}$ and *lrpprc*$^{GBT0235/GBT0235}$ bands.

## Genotyping of zebrafish larvae by fin clipping

This larval genotyping protocol was adapted from the previously described study (*Wilkinson et al., 2017*). Larvae were placed in a petri dish containing 3 µL of 20% Tween-20 per 30 mL of embryo media. Larvae were individually placed on an inverted petri dish under a dissecting scope using transillumination. A scalpel was used to cut the tail distal to the blood flow. The fin clip was removed from the solution using forceps and placed into a PCR strip tube containing 10 µL of 50 mM NaOH. Scalpel and forceps were sterilized with an ethanol wipe between biopsies. Larvae were transferred into a petri dish with fresh embryo media before being moved to a 24-well plate for recovery while genotyping was performed. Tail biopsies were capped and heated to 98 °C for 10 min and then cooled to 25 °C. 10 µL of 100 mM Tris pH 8 was added to each tube and mixed. These samples were then used

for PCR amplification. Following genotyping, embryos were grouped by genotype and placed in petri dishes until needed for experiments.

## Non-invasive enzymatic genotyping

To assess the genotype of the larvae, a non-invasive method of genotyping was adapted with modifications (*Zhang et al., 2020*; *Sabharwal et al., 2020*). Larvae aged 4 dpf were washed in embryo media and then briefly rinsed in collection buffer (1 X tricaine solution, 30 mM Tris-HCl). Larvae were transferred to a fresh petri dish and collection buffer was aspirated and replaced with genotyping buffer (25 µg/ml proteinase K, 30 mM Tris-HCl, 1 X tricaine). Individual larvae were aspirated in 40 µL of genotyping buffer and transferred to the individual well of a 96-well plate. The plate was incubated in a shaker at 37 °C for 20 min at 500 RPM. Following incubation, 25 µL of the solution was transferred to fresh PCR tubes containing 15 µL of the lysis buffer (10 mM Tris-HCl, pH 8.0, 50 mM KCl, 0.3% Tween-20, 0.3% NP-40, and 1 mM EDTA). The solution mix was incubated at 98 °C for 5 min to deactivate proteinase K. The DNA obtained was used as a template for the genotyping PCR. Larvae were then rescued from the remaining 15 µL of the genotyping buffer and rinsed in fresh embryo water.

## Capturing the spatio-temporal expression dynamics in the *in vivo* protein trap lines

Larvae were treated with 0.2 mM phenylthiocarbamide (PTU) at 24 hours post-fertilization (hpf) to prevent pigmentation. Fish were anesthetized with 1 X tricaine and mounted in 1.5% agarose (Fisher Scientific, USA) prepared with 1 X tricaine solution (0.18 mg/L) in an agarose column in the imaging chamber. For Lightsheet microscopy, larval zebrafish were anesthetized with 1 X tricaine in embryo water during the imaging procedure. RFP expression patterns of 6 dpf larval zebrafish were captured using LP 560 nm filter as excitation and LP 585 nm as emission using a Lightsheet 7 microscope (Zeiss, Germany) with 5 X objective. Confocal microscopy was carried out by embedding the 6 dpf larvae in 1% (w/v in E2 medium containing tricaine) low-melting agarose (Fisher Scientific, USA) in a 50 mm glass-bottom dish (MatTek, USA). Once the agarose was set, it was immersed in an E2 medium containing tricaine to prevent drying. Each set of brightfield and RFP images for both heterozygous and homozygous mutants were acquired using Zeiss LSM780 (63XW/1.2 NA) and the images shown are the maximum image projections of the z-stacks obtained from each direction. Laterally oriented Z-stacks of representative images of the caudal fin for GFP and mRFP expressions were captured using the laser with Ex488 and Ex561, respectively under a 63XW/1.2 NA water objective. Each set of images was independently acquired at different fluorescent wavelengths and a composite image was generated using Zeiss Zen microscope software. For subcellular localization, *lrpprc*GBT0235/+ adult zebrafish were outcrossed to *Tg(MLS:EGFP)* zebrafish (*Kim et al., 2008*; *Sabharwal et al., 2019b*) to obtain *Tg(MLS:EGFP) lrpprc* GBT0235/+. Embryos were maintained in 0.2 mM phenylthiocarbamide from 24 hpf and anesthetized with tricaine (0.18 mg/L). For *Figure 3D*, subcellular localization was studied by injecting the Tag BFP RNA with the nuclear localization signal in the embryos obtained from the above cross. Sequence-verified pT3TS-NLS:BFP clones were linearized with the PstI restriction enzyme. For *in vitro* transcription reaction using the T3 mMessage mMachine transcription kit (Thermo Scientific, USA), the linearized plasmid was used as a template. mRNA was purified using the NEB Monarch RNA cleanup kit (NEB, USA). A total of 50 pg of mRNA was injected into single-cell zebrafish embryos obtained from the outcross of *Tg(MLS:EGFP)* and *lrpprc*GBT0235/+. Injected embryos were then raised in embryo water at 29 °C. At the time of imaging, embryos were anesthetized with tricaine (0.18 mg/L) and embedded in 1% low melting agarose as described in the previous section. Embryos were viewed under a Zeiss LSM780 confocal microscope. Laterally oriented Z-stacks of representative images for GFP and mRFP expressions were captured using the laser with Ex488 and Ex561 and TagBFP was captured using the laser for DAPI, respectively under a 63XW/1.2 NA water objective. Each set of images was independently acquired at different fluorescent wavelengths and a composite image was generated using Zeiss ZEN microscope software.

## Survival assay

*lrpprc* GBT0235/+ adult zebrafish were crossed with *Tg(−2.8fabp10:Cre;−0.8cryaa:Venus)*S955 to obtain double-transgenic zebrafish expressing both GBT cassette and the *fabp10* driven *Cre* recombinase. Larval zebrafish were screened for RFP and gamma crystalline Venus (V) and RFP+/V- group were

sorted based on the presence of darkened liver (DL) phenotype on a brightfield microscope. Live counts were recorded daily from 6 dpf to 12 dpf and larvae from each group were counted and euthanized for genotyping using NaOH extraction, followed by PCR using primers listed in *Supplementary file 1A*.

## RNA extraction and sample preparation

*lrpprc $^{GBT0235/+}$* adult zebrafish were incrossed and larvae were genotyped at 6 dpf. RNA isolation was performed using TRIzol extraction (Qiagen, USA). For RNA sequencing and qPCR, 6 and 2 genotyped larvae respectively, were placed in 500 µL of TRIzol and homogenized using a handheld homogenizer on ice and then incubated at room temperature for 5 min. Chloroform was added followed by 15-min incubation on ice before spinning at 12,000 g at 4 °C for 15 min. The aqueous layer was removed and placed into a separate 1.7 mL centrifuge tube and 1 X volume of isopropanol was added, followed by 1 hr incubation at –20 °C to precipitate RNA. The pellet was washed with 70% ethanol and resuspended in nuclease-free water. It was then subjected to DNAse treatment to isolate and purify the RNA. RNA quantity was estimated using the spectrophotometer.

## Mitochondrial DNA relative gene expression analysis

A total of 300 ng of RNA was used for reverse transcription into cDNA using the SuperScript III kit (Thermo Fisher Scientific, USA). qPCR was performed using the SensiFAST SYBR Lo-ROX kit (Bioline, USA). Wild-type and mutant transcripts levels were normalized to eukaryotic translation elongation factor 1 alpha 1, like 1 (*eef1a1l1*). Primer sequences are provided in *Supplementary file 1A*.

## Relative assessment of mtDNA copy number

Relative mitochondrial copy number across the homozygous mutants and wild-type siblings was estimated using total genomic DNA extracted by the following procedure. Individual embryos were incubated in DNA extraction buffer (50 mM Tris– HCl pH 8.5, 1 mM EDTA, 0.5% Tween-20, 200 µg/ml proteinase K) at 55 °C overnight. Following incubation, the lysate was centrifuged at maximum speed for 2 min and the supernatant was transferred to the fresh PCR tube. Relative mtDNA abundance was measured using primers for *polg* as the nuclear control and *mt-nd1* as the mitochondrial control. mtDNA copy number PCR was performed using the Sensifast SYBR Lo-ROX kit (Bioline, USA).

## Lactate measurement in zebrafish embryos

*lrpprc $^{GBT0235/+}$* zebrafish were incrossed and sorted as per genotype. Seven dpf zebrafish larvae were pooled together and lysed in 100 µL of 1 X PBS. 100 µL of lysate was spiked with a mixture of labeled internal standards following PFB-oximation of keto acids. The samples were acidified with 6 N HCl and NaCl and extracted into ethyl acetate. After evaporation, the dry residue was silylated with N,O,-bis-(trimethylsilyl) trifluoroacetamide with 1% trimethylchlorosilane (BSTFA + TMCS). Subsequently, the reaction mixture was analyzed by capillary gas chromatography/mass spectrometry (GC/MS) (GC: Agilent 7890B Gas Chromatograph; mass spectrometer is an Agilent 5977 A Mass Spectrometer, EI Source, and ionization) using selected ion monitoring (SIM) with positive electron impact ionization (EI) and stable isotope dilution. Peaks were acquired using Agilent Masshunter GC/MS Acquisition 10.0.368 and the software used for quantitation was Agilent Quantitative Analysis Masshunter Version 10.0.

## Birefringence assay

*lrpprc $^{GBT0235/+}$* adult zebrafish were in-crossed to obtain a population of *lrpprc $^{+/+}$*, *lrpprc $^{GBT0235/+}$*, and *lrpprc $^{GBT0235/GBT0235}$* offspring. *Tg(fabp10:Cre)/lrpprc $^{GBT0235/+}$* were incrossed to obtain *Tg(fabp10:Cre)/lrpprc $^{GBT0235/235}$*, homozygous mutants only expressing Cre recombinase in the liver. Embryos were maintained in an E3 embryo medium at 28.5 °C at a density of 30 per dish and kept on a 14 hr light/10 hr dark cycle. At 4 dpf, larvae were anesthetized with tricaine (0.18 mg/L) and embedded in 1% low melting agarose as described in the previous section. The larvae were viewed using a Stemi SV11 stereomicroscope (Zeiss, Germany) equipped with a polarized lens underneath the stage and a polarized lens on the objective lens (*Smith et al., 2013*). The polarized lens on the objective was rotated until the background went dark. Birefringence images were then acquired using a DSLR camera (Canon, Powershot G10) in RAW mode using an ISO of 200, aperture of f/4.5, and an exposure

of 1/15 s. Larvae were subsequently rescued from agarose using ultra-fine forceps, placed into 8-strip tubes, euthanized by tricaine overdose, and lysed for DNA extraction using 30 µL of 50 mM NaOH (incubated at 95 °C for 25 min, then neutralized with 3 µL of 1 M Tris-HCl (pH 8)). This neutralized lysate was used for PCR genotyping.

RAW images were converted to 8-bit grayscale TIFFs in Adobe Photoshop. Grayscale TIFF birefringence images were imported into FIJI for further analysis. We outlined an area in each fish that corresponded to the birefringence from its body using the wand tool (legacy mode, threshold = 16). The birefringence signal from the otolith was specifically excluded from these measurements. All outlined regions were saved as regions of interest (ROIs). Within these ROIs, measurements for area, mean gray value, min gray value, max gray value, and integrated gray value density were obtained. During data collection and analysis, experimenters were blind to the exact genotype and only knew the RFP expression pattern of the larvae. After un-masking the data, area, mean gray value, and integrated gray value density were utilized to make plots and assessed for statistical differences between $lrpprc^{+/+}$, $lrpprc^{GBT0235/GBT0235}$, and $Tg(fabp10:Cre)/lrpprc^{GBT0235/235}$, animals. Each data point represents a single animal.

## Acridine orange staining

Larvae were stained with 1 ml of 100 µg/ml of acridine orange (Sigma Aldrich, USA) stain dissolved in the E3 embryo medium and incubated in dark for 30 minutes. The larvae were viewed using an Axiovision fluorescence stereomicroscope (Zeiss, Germany). CZI files were exported as TIFF files in ZEN software and image file names were de-identified among genotypes. Images were then imported into FIJI and converted into 8-bit grayscale for particle analysis. Thresholding was performed with 41 as the lower limit and 87 as the upper limit to only visualize particles. The analyze particles tool was used to measure particles of all sizes and roundness. ROIs and results for each image were saved as a CSV file. After re-identifying the image file names, the particle count was used to compare genotypes by making plots in GraphPad Prism. Statistical differences between particle numbers for $lrpprc^{GBT0235/GBT0235}$ and $lrpprc^{+/+}$ were assessed using an unpaired t-test.

## RNA sequencing and analyses

RNA library preparations and sequencing reactions were performed at GENEWIZ, LLC. (USA). Six individual larvae from each genotype were pooled for RNA isolation for transcriptome sequencing. Quantification of RNA samples was carried out using Qubit 2.0 Fluorometer (Thermo Fisher Scientific, USA) and further, the RNA integrity was checked using Agilent Tapestation 4200 (Agilent Technologies, USA). RNA sequencing libraries were prepared using the NEBNext Ultra RNA Library Prep Kit for Illumina using the manufacturer's instructions (NEB, USA). Briefly, mRNAs were initially enriched with oligo(dT) beads. Enriched mRNAs were fragmented for 15 min at 94 °C. Subsequently, cDNA fragments were synthesized, end-repaired, and adenylated at 3'ends, and universal adapters were ligated to them, followed by index addition and library enrichment by PCR with limited cycles. The sequencing library was validated on the Agilent TapeStation (Agilent Technologies, USA), and quantified by using Qubit 2.0 Fluorometer (Thermo Fisher Scientific, USA) as well as by quantitative PCR (KAPA Biosystems, USA). The sequencing libraries were clustered on a single lane of a flow cell. After clustering, the flow cell was loaded on the Illumina HiSeq4000 instrument according to the manufacturer's instructions. The samples were sequenced using a 2x150 bp Paired-End (PE) configuration. Image analysis and base calling were conducted by the HiSeq Control Software (HCS). Raw sequence data (.bcl files) generated from Illumina HiSeq was converted into fastq files and de-multiplexed using Illumina's bcl2fastq 2.17 software. One mismatch was allowed for index sequence identification.

Paired-end reads of 150 bp length generated for the $lrpprc^{GBT0235/GBT0235}$ and $lrpprc^{+/+}$ were trimmed with a base quality cut-off of Phred score Q30 using Trimmomatic (**Bolger et al., 2014**). Filtered reads were pseudoaligned to the zebrafish reference genome Zv10 version. Kallisto (**Bray et al., 2016**) was used for transcript assembly and calculation of relative expression values across transcripts. Tximport (**Soneson et al., 2015**) was used to summarize transcript level counts at the gene level. Differential expression was computed using DESeq2 (**Love et al., 2014**). Genes with a fold change of ≥ $\log_2 1$ and ≤ $\log_2 -1$ and adjusted p-value <0.05 were considered to be upregulated and downregulated respectively.

## Identification of differentially expressed zebrafish orthologs for Mitocarta genes

Human orthologs of differentially expressed zebrafish genes were identified from ZFIN (Zebrafish Information Network) and Ensembl. Differentially expressed human orthologs of zebrafish genes were run against the human Mitocarta 3.0 database (*Rath et al., 2021*) to identify upregulated and down-regulated mitochondrial genes involved in biological pathways resident in mitochondria. Differentially expressed genes were entered in the PANTHER (*Mi et al., 2019*) database to identify the protein class and biological process of these genes. For the differentially expressed genes, we predicted the enriched or depleted pathways by using a web-based integrated analysis platform, GeneTrail2 (*Stöckel et al., 2016*). Differentially expressed genes along with their respective calculated scores were fed as the input file. Scores were calculated as per the following formula: Score = [-$\log_{10}$(adjusted p-value) * $\log_2$fold change]. Kolmogorov-Smirnov test was used as the statistical test for the gene set enrichment analysis to find enriched/depleted biological pathways from the KEGG pathways, Reactome pathways, and Wiki pathways.

## Oil Red O staining

For whole-mount staining at the larval stage, 8 dpf larvae were fixed in 4% paraformaldehyde overnight. The Oil Red O staining procedure was followed as described previously (*Kim et al., 2013*). Briefly, $lrpprc^{+/+}$, $lrpprc^{GBT0235/GBT0235}$ and $Tg(fabp10:Cre)lrpprc^{GBT0235/GBT0235}$ larvae were rinsed three times (5 min each) with 1×PBS/0.5% Tween-20 (PBS-T). After removing PBS-T, larvae were stained with a mixture of 300 µL of 0.5% ORO in 100% isopropyl alcohol and 200 µL of distilled water for 15 min. Larvae were then rinsed with 1×PBS-Tween-20 three times, twice in 60% isopropyl alcohol for 5 min each, and then mounted for imaging. Oil Red O images were captured with a color camera in JPEG format. For each batch of images (corresponding to one day of experiments), an image with dark Oil Red O staining and an image with light Oil Red O staining were respectively used to determine the lower and upper threshold values for positive staining. Briefly, the lower edge of the threshold was set as the level that most pixels representing melanocytes were darker than, but almost all dark Oil Red O staining was brighter than. Similarly, the upper edge of the threshold was determined to be at the level where most pixels representing the background outside of the fish were brighter than, but some light Oil Red O staining was darker than. This left a range between upper and lower threshold limits of 40–60 values that encompassed the most Oil Red O staining. To quantify only the Oil Red O staining in the liver, we drew a single region of interest that bordered the liver of the most darkly stained larvae and used this same ROI template between replicates. All outlined regions were saved as regions of interest (ROIs). Within these ROIs, measurements for the area were obtained and plots and statistical analyses were performed. This region of interest (ROI) specifically excluded the area around the swim bladder due to the swim bladder having high amounts of background stain in the majority of images. Within this ROI, the area of pixel values within the threshold in each larva was determined using the 'limit to threshold' option in the 'set measurements' menu in FIJI and normalized to the average oil red stain area of the wild-type group.

## Electron microscopy of hepatocyte mitochondria

Zebrafish larvae were fixed and imaged using transmission electron microscopy as described in the previous study (*Wilson et al., 2020*). Briefly, 8 dpf embryos were fixed for 1–3 hr in a mixture of 3% glutaraldehyde, 1% formaldehyde, and 0.1 M cacodylate. Embryos were embedded in 2% low melt agarose followed by processing as described earlier (*Zeituni et al., 2016*). Following embedding steps, post-fixation for 1 hr with 1% osmium tetroxide and 1.25% potassium ferricyanide in cacodylate solution occurred. Larvae were rinsed twice with water and incubated in 0.05 maleate pH 6.5 before staining overnight at 4 °C in 0.5% uranyl acetate. Following overnight incubation, samples were washed with water and dehydrated using ethanol dilution. Samples were washed with propylene oxide and then incubated with propylene oxide/resin followed by an overnight evaporation period. Finally, larvae were embedded in 100% resin at 55 °C overnight and then 70 °C for three days. Sectioning was performed on Reichert Ultracut-S (Leica Microsystems, Germany), mounted on mesh grids, and stained with lead citrate. Images were taken using a Phillips Technai-12 electron microscope and a 794 Gatan MultiScan CCD camera. Electron micrographs were obtained from larvae from each of the genotypes: $lrpprc^{+/+}$, $lrpprc^{GBT0235/GBT0235}$, and $Tg(fabp10:Cre)/lrpprc^{GBT0235/GBT0235}$.

## Feeding of fluorescently tagged long-chain fatty acids

In each of the four experiments, an equal number of 6 dpf larvae of each genotype (N=18-24) was pooled and arrayed separately in Falcon brand 6-well tissue culture plates for delivery of a fluorescently tagged long-chain fatty acid feed. 4,4-difluoro-5,7-dimethyl-4-bora-3a,4a-diaza-*s*-indacene 3-dodecanoic acid (BODIPY FL C12; Thermo Fisher Scientific, USA) (4 µg/mL) was emulsified in a solution of 5% chicken egg yolk liposomes (in embryo medium) as previously described (*Carten et al., 2011*). Since larvae can vary greatly in the amount they ingest, non-metabolizable TopFluor Cholesterol (conc, 23-(dipyrrometheneboron difluoride)–24-norcholesterol; Avanti Lipids, USA) was added to the feed solution to serve as a readout of the amount ingested, allowing for normalization of the HPLC output. TopFluor Cholesterol is harder to get into solution compared to BODIPY FL C12: embryo medium and 1% fatty acid-free BSA was prewarmed to 30 °C. After 4 hr of feeding in a light-shielded shaking incubator (30 °C, 30 rpm), larvae were rinsed in fresh embryo media and screened for ingestion (darkened intestines) under a stereomicroscope. Larvae that did not eat were excluded from the study. Larvae that ate were moved to tissue culture plates with fresh embryo media for an overnight chase period of 12.5 hr in a 28.5 °C light-cycling (14 hr on, 10 hr off) incubator. Pools of 8–13 larvae from each group were snap-frozen on dry ice in microcentrifuge tubes and stored dry at –80 °C for lipid extraction.

## HPLC analyses of long-chain fatty acids

Frozen samples of pooled larvae were subjected to lipid extraction using a modified Bligh-Dyer procedure. Extractions were dried and resuspended in HPLC-grade isopropanol. Sample components were separated and detected by an HPLC system as described previously (*Quinlivan et al., 2017*). Chromatographic peak baselines were manually delimited and peak areas were automatically measured (Chromeleon 7.2; Thermo Fisher Scientific). Peak areas per larval equivalent were calculated and normalized to the TopFluor Cholesterol peak (non-metabolizable; added to the fluorescent fatty acid feed). To fit a linear model wherein the outcomes may have possible unknown correlations, generalized estimated equations (gee) were performed on the R platform comparing total non polar lipid area as well as individual peaks/groups of peaks. Assessment of normality by visual examination of Q-Q (quantile-quantile) plots confirmed that the standard normal curve is a valid approximation for the distribution of the data. P-values were obtained from the standard normal Z-table.

## Statistical analyses

Plots and statistical analyses including the Mann-Whitney test, unpaired t-test, and construction of heat map were performed in GraphPad Prism 8. Kolmogorov-Smirnov test was used as the statistical test for the gene set enrichment analysis on the GeneTrail webserver. Results of the statistical analyses have been described either in the methods, results section, or in the figure legends.

## Acknowledgements

The authors thank Ms. Mrunal Dehankar of the Mayo Clinic Bioinformatics core for the assistance with bioinformatics analyses, the Mayo Clinic Zebrafish Facility staff, and the Mayo Clinic Microscopy Cell Analysis and Core for the imaging facility. We would also like to acknowledge Mike Sepanski and Vanessa Quinlivan of the Carnegie Institution for the TEM imaging and assistance with HPLC, respectively. *Tg(−2.8fabp10:Cre;−0.8cryaa:Venus)*[S955] zebrafish transgenic line was a kind gift from Dr. W Chen, Vanderbilt University, TN, USA. *Tg(MLS:EGFP)* zebrafish transgenic line was a kind gift from Dr. S Sivasubbu, CSIR-IGIB, India, provided by Dr. S Choi, Chonnam Medical School, South Korea. We would also like to thank Dr. Gaofeng Cui, Mayo Clinic for the assistance in setting up with RoseTTA-Fold server. This work was funded by NIH grants GM63904 and DA14546, the Marriott Foundation, the Mayo Foundation, and R01 DK093399 (Farber, PI). Additional support for this work was provided by the Carnegie Institution for Science Endowment and the G Harold and Leila Y Mathers Charitable Foundation.

## Additional information

### Funding

| Funder | Grant reference number | Author |
|---|---|---|
| National Institutes of Health | GM63904 | Stephen C Ekker |
| National Institutes of Health | DA14546 | Stephen C Ekker |
| Marriott Foundation | | Stephen C Ekker |
| Mayo Foundation for Medical Education and Research | | Stephen C Ekker |
| National Institutes of Health | DK093399 | Steven A Farber |
| Carnegie Institution for Science | | Steven A Farber |
| G. Harold and Leila Y. Mathers Charitable Foundation | | Steven A Farber |

The funders had no role in study design, data collection and interpretation, or the decision to submit the work for publication.

### Author contributions

Ankit Sabharwal, Conceptualization, Data curation, Formal analysis, Validation, Investigation, Visualization, Methodology, Writing - original draft, Writing - review and editing; Mark D Wishman, Conceptualization, Data curation, Formal analysis, Validation, Investigation, Visualization, Methodology, Writing - original draft; Roberto Lopez Cervera, Conceptualization, Data curation, Validation, Investigation, Visualization, Methodology; MaKayla R Serres, Data curation, Validation, Investigation, Methodology; Jennifer L Anderson, Data curation, Formal analysis, Validation, Investigation, Visualization, Methodology, Writing - original draft; Shannon R Holmberg, Jean M Lacey, William J Laxen, Perry R Loken, Investigation, Methodology; Bibekananda Kar, Investigation, Methodology, Writing - original draft, Writing - review and editing; Anthony J Treichel, Noriko Ichino, Validation, Investigation, Visualization, Methodology; Weibin Liu, Jingchun Yang, Yun Deng, Validation, Investigation, Methodology; Yonghe Ding, Investigation, Visualization, Methodology; Devin Oglesbee, Formal analysis, Investigation, Methodology; Steven A Farber, Resources, Supervision, Funding acquisition, Investigation, Project administration; Karl J Clark, Conceptualization, Resources, Supervision, Funding acquisition, Investigation, Visualization, Methodology, Project administration; Xiaolei Xu, Formal analysis, Supervision, Investigation, Methodology; Stephen C Ekker, Conceptualization, Resources, Formal analysis, Supervision, Funding acquisition, Investigation, Methodology, Writing - original draft, Project administration, Writing - review and editing

### Author ORCIDs

Ankit Sabharwal (ID) http://orcid.org/0000-0003-4355-0355
Anthony J Treichel (ID) http://orcid.org/0000-0002-4393-7034
Noriko Ichino (ID) http://orcid.org/0000-0002-7009-8299
Steven A Farber (ID) http://orcid.org/0000-0002-8037-7312
Karl J Clark (ID) http://orcid.org/0000-0002-9637-0967
Xiaolei Xu (ID) http://orcid.org/0000-0002-4928-3422
Stephen C Ekker (ID) http://orcid.org/0000-0003-0726-4212

### Ethics

All adult zebrafish and embryos were maintained according to the guidelines established by Mayo Clinic Institutional Animal Care and Use Committee (IACUC number: A34513-13-R16).

### Decision letter and Author response

Decision letter https://doi.org/10.7554/eLife.65488.sa1

Author response https://doi.org/10.7554/eLife.65488.sa2

## Additional files

### Supplementary files

• Supplementary file 1. Supplementary tables. (A) List of oligonucleotides used in the study. (B) Prediction of Human LRPPRC binding sites within the human mitochondrial genome (NC_012920.1). The top twenty matches among both strands of the complete mitochondrial genome are shown. The p-values were calculated with the FIMO program and were used for ranking. (C) Prediction of zebrafish Lrpprc binding sites within the zebrafish mitochondrial genome (NC_002333.2). The top twenty matches among both strands of the complete mitochondrial genome are shown. The p-values were calculated with the FIMO program and were used for ranking.

• Supplementary file 2. Gene set enrichment analysis for *lrpprc* homozygous mutants.

• Transparent reporting form

### Data availability

All data generated or analysed during this study are included in the manuscript and supporting files. Source data is provided along with the manuscript. Raw sequencing data has been uploaded on NCBI SRA (PRJNA683704).

The following dataset was generated:

| Author(s) | Year | Dataset title | Dataset URL | Database and Identifier |
|---|---|---|---|---|
| Sabharwal A, Wishman MD, Cervera RL, Serres MR, Anderson JL, Treichel AJ, Ichino N, Liu W, Yang J, Ding Y, Deng Y, Farber SA, Clark KJ, Xu X, Ekker SC | 2020 | Zebrafish Model for LSFC | https://www.ncbi.nlm. nih.gov/bioproject/ PRJNA683704/ | NCBI Sequence Read Archive, PRJNA683704 |

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
