## [Editor Report]

This study presents an interesting discovery that reversion of the mutation in the liver largely rescues the lethality in a zebrafish model of Leigh Syndrome, French Canadian Type (LSFC). The evidence is strong as 90% of the liver revertants survive past the terminal age. This manuscript is of interest to scientists in the field of mitochondrial biology and pathology.

---

## [Decision Letter]

**Decision letter after peer review:**

Thank you for sending your article entitled "A Genetic Model Therapy Proposes a Critical Role for Liver Dysfunction in Mitochondrial Biology and Disease" for peer review at *eLife*. Your article is being evaluated by 2 peer reviewers, one of whom is a member of our Board of Reviewing Editors, and the evaluation is being overseen by Didier Stainier as the Senior Editor.

*Reviewer #1 (Recommendations for the authors):*

The manuscript by Sabharwal et al. describes a zebrafish model of Leigh Syndrome, French Canadian Type (LSFC). The authors further characterized a mutant line initially reported in a previous *eLife* resources paper that describe a collection of mutations that reveal the expression dynamics of mutated gene and can be reverted by Cre. They showed that an insertion of GBT into *Lrpprc* gene, whose human ortholog underlies the more rare form of LSFC, disrupts the gene product, leading to hallmarks of LSFC including fatty liver, skeletal muscle defects, and death by 11 dpf in the homozygotes. This thus represents a zebrafish model of LSFC. As expected, the gene product is localized to mitochondria. Further analyses demonstrated a decrease of mitochondrial transcripts, an increase of non-polar lipids, and altered expression of genes associated with lipid metabolism and peroxisome associated pathways. Taking advantage of the reversibility of the mutation, the authors further show that correction of the mutation only in the liver is sufficient to largely correct the lethality and the lipid profile. Although brain and liver mitochondrial function of *Lrpprc* is known to more severely impacted than other tissues in the patients due to their high metabolic demand, that rescue of *Lrpprc* function in liver can largely reverse the lethality is an interesting and unexpected finding. The studies further demonstrate the advantages of the mutant collection. Although the manuscript is clearly written and data are of good quality, several issues in the current form need to be addressed.

1) Much of the analysis focuses on known phenotypes in lipid metabolism of *Lrpprc* deficiency in mammals. LSFC patients have defects in many tissues/organs, e.g. neuronal necrosis. These are at least partially independent of liver dysfunction. Does the liver specific *Lrpprc* rescue affect neuronal death?

2) The product of *Lrpprc* is important for several aspects of mitochondrial biology including biogenesis. The authors did not address mitochondrial copy number. Do mutant cells have fewer mitochondria? This is also important in interpreting the qRT-PCR results (Figure 3B). Is the decreased expression due to a decrease of mitochondrial number, RNA stability, or both?

*Reviewer #3 (Recommendations for the authors):*

In this manuscript, the authors aim to create a zebrafish model for Leigh Syndrome French Canadian Type (LSFC) by knocking out the irpprc gene responsible for the disease. In previous works, the authors collected transgenic fish lines that labelled specific proteins via a protein trap approach and identified the *Lrpprc* knockout fish. The authors analyzed the homozygous mutant phenotypes by RNA-seq, microscopic analyses, electron microscopy, and a fluorescent fatty acid analog. The authors found deficits in liver and muscle development and in liver functions (including lipid metabolism). Further, taking advantage of the feature of the protein trap vector, the orientation of the protein trap vector was reversed and the mutant phenotypes were rescued. Thus, by using a useful model vertebrate zebrafish, the authors successfully established a conditionally controllable LSFC model, which should be useful to understand the function of the *Lrpprc* gene and the mechanism of the disease.

The major strength of this work is (1) detailed descriptions of the mutant phenotypes (lipid accumulation in the liver, mitochondrial morphology defects, and altered lipid metabolism), and (2) successful demonstration of genetic rescue of the mutant phenotypes.

The weakness of the work is (1) a mouse model in which the *Lrpprc* gene was knocked out specifically in the hepatocytes has been described previously, (2) overall the manuscript was rather descriptive and poor in mechanistic aspects of the *Lrpprc* function and pathology of LSFC, and (3) it was not clearly shown how this genetic model is useful to study the function of *Lrpprc* and open a way to new therapeutic options as the authors discussed in the Discussion section.

1) In abstract, the authors say, "created 1200 transgenic zebrafish strains tagging protein-coding genes". However, in their previous manuscript (Ichino et al. 2020), out of 1200, 204 lines were proved to actually trap proteins. Therefore, this number should be 204.

2) The mechanistic analysis of muscle defects was lacking.

3) The mechanism how down-regulated and upregulated genes in the mutants are controlled by *Lrpprc* was not experimentally analyzed and described.

4) How was the mitochondria morphology altered was not shown.

5) The mechanism to control lipid metabolism was totally unknown.

---

## [Author Response]

Reviewer #1 (Recommendations for the authors):The manuscript by Sabharwal et al. describes a zebrafish model of Leigh Syndrome, French Canadian Type (LSFC). The authors further characterized a mutant line initially reported in a previous eLife resources paper that describe a collection of mutations that reveal the expression dynamics of mutated gene and can be reverted by Cre. They showed that an insertion of GBT into Lrpprc gene, whose human ortholog underlies the more rare form of LSFC, disrupts the gene product, leading to hallmarks of LSFC including fatty liver, skeletal muscle defects, and death by 11 dpf in the homozygotes. This thus represents a zebrafish model of LSFC. As expected, the gene product is localized to mitochondria. Further analyses demonstrated a decrease of mitochondrial transcripts, an increase of non-polar lipids, and altered expression of genes associated with lipid metabolism and peroxisome associated pathways. Taking advantage of the reversibility of the mutation, the authors further show that correction of the mutation only in the liver is sufficient to largely correct the lethality and the lipid profile. Although brain and liver mitochondrial function of Lrpprc is known to more severely impacted than other tissues in the patients due to their high metabolic demand, that rescue of Lrpprc function in liver can largely reverse the lethality is an interesting and unexpected finding. The studies further demonstrate the advantages of the mutant collection. Although the manuscript is clearly written and data are of good quality, several issues in the current form need to be addressed.1) Much of the analysis focuses on known phenotypes in lipid metabolism of Lrpprc deficiency in mammals. LSFC patients have defects in many tissues/organs, e.g. neuronal necrosis. These are at least partially independent of liver dysfunction. Does the liver specific Lrpprc rescue affect neuronal death?

Authors appreciate the suggestion provided by the reviewer. Reviewers have brought up an interesting aspect of exploring the rescue of other tissue defects such as brain or muscle upon the reversion of *Lrpprc* expression in the liver. We did experiments to assess the muscle phenotype by birefringence microscopy and neural necrosis by staining the larvae for acridine orange.

In addition to the above comment, we were intrigued to see whether the muscle phenotype is dependent on the defects in the liver lipid homeostasis. To see the global effect of liver-specific rescue, we investigated the muscle phenotype in the liver rescued *Lrpprc* homozygous larvae, *Tg(fabp10:Cre)/Lrpprc^GBT0235/GBT0235^*. We measured muscle birefringence of the mutants at 4-days post-fertilization as an assessment of overall muscle structure. The analysis from the muscle studies on mutants revealed that *Lrpprc* homozygous larvae displayed a lower birefringence signal highlighting a disrupted muscle structure/integrity. A similar phenotype was also observed in liver-rescued homozygous animals, where muscle integrity was not observed to be restored (data provided below). These results possibly highlight that the muscle defects do not contribute to the larval lethality that we see in the loss of function embryos and the liver-specific rescue wasn’t able to restore the muscle phenotype, despite the liver-rescued rescued larvae showing improved survival.

We have modified the text and included the above findings in the revised manuscript:

Results (Page 8; lines 194-197): “Assessment for the muscle phenotype in the 4 dpf liver-rescued homozygous mutants displayed a similar decrease in the parameters for muscle integrity such as integrated birefringence density as compared to homozygous mutants (Figure 5—figure supplement 1).”

Discussion (Page 16; lines 421-425): “Interestingly, muscle phenotypes that were displayed by *Lrpprc* homozygous mutants were not mitigated by the liver-specific reversion of the mutagenicity cassette. This not only underscores the specificity of the revertible cassette in a given tissue but also indicates the improved survival and lipid homeostasis observed in rescued animals being at least partially independent of muscle dysfunction.”

Materials and methods (Page 32; lines 694-696): “*Tg(fabp10:Cre)/Lrpprc^GBT0235/+^* were incrossed to obtain *Tg(fabp10:Cre)/Lrpprc^GBT0235/235^*, homozygous mutants only expressing Cre recombinase in the liver.”

To address the question if the liver specifically affects neuronal death, the authors looked for neuronal death first in the *Lrpprc* homozygous mutants. To observe any incidence of neuronal necrosis, *Lrpprc* homozygous mutants and wild-type embryos were stained with acridine orange. Findings from the study didn’t reveal any appreciable enriched neuronal necrosis in the homozygous embryos. In the graph provided below, we did an unbiased analysis on blinded images, wherein we counted the number of spots representing acridine orange (AO) stain across both genotypes. The difference in the number of AO spots was not observed to be significantly different across both genotypes.

We have modified the text and included the above findings in the revised manuscript:

Results (Page 8; lines 198-202): “A subset of patients also displays neural phenotypes such as lesions in the brainstem and basal ganglia. To investigate for any evidence of neural necrosis, acridine orange (AO) staining was performed on *Lrpprc* homozygous animals and wild-type siblings. Performing a blindfold analysis on the representative images across both genotypes, we didn’t notice any significant difference in the necrotic signal represented by acridine orange spots (Figure 5—figure supplement 2).”

Materials and methods (Pages 32-33; lines 718-729): “Acridine orange staining – Larvae were stained with 1 ml of 100 µg/ml of acridine orange (Σ Aldrich, USA) stain dissolved in the E3 embryo medium and incubated in dark for 30 minutes. The larvae were viewed using an Axiovision fluorescence stereomicroscope (Zeiss, Germany). CZI files were exported as TIFF files in ZEN software and image file names were de-identified among genotypes. Images were then imported into FIJI and converted into 8-bit grayscale for particle analysis. Thresholding was performed with 41 as the lower limit and 87 as the upper limit to only visualize particles. The analyze particles tool was used to measure particles of all sizes and roundness. ROIs and results for each image were saved as a CSV file. After re-identifying the image file names, the particle count was used to compare genotypes by making plots in GraphPad Prism. Statistical differences between particle numbers for *Lrpprc^GBT0235/GBT0235^* and *Lrpprc^+/+^* were assessed using an unpaired t-test.”

2) The product of Lrpprc is important for several aspects of mitochondrial biology including biogenesis. The authors did not address mitochondrial copy number. Do mutant cells have fewer mitochondria? This is also important in interpreting the qRT-PCR results (Figure 3B). Is the decreased expression due to a decrease of mitochondrial number, RNA stability, or both?

We are thankful to the reviewer for their suggestion. We did investigate the mitochondrial copy number and observed that there was no significant decrease or increase in the mtDNA copy number across both genotypic classes. These results do provide insights into the decreased transcript levels not being contributed by the mitochondrial DNA depletion. These findings are consistent with the published studies where the authors also didn’t observe any change in the mtDNA copy number in the cellular models mimicking the loss of function of *LRPPRC* (Gohil et al., 2010).

We have modified the text and included the above findings in the revised manuscript:

Materials and methods (Page 31; lines 671-679): “Relative assessment of mtDNA copy number – Relative mitochondrial copy number across the homozygous mutants and wild-type siblings was estimated using total genomic DNA extracted by the following procedure. Individual embryos were incubated in DNA extraction buffer (50 mM Tris– HCl pH 8.5, 1 mM EDTA, 0.5% Tween-20, 200 μg/ml proteinase K) at 55°C overnight. Following incubation, the lysate was centrifuged at maximum speed for 2 minutes and the supernatant was transferred to the fresh PCR tube. Relative mtDNA abundance was measured using primers for polg as the nuclear control and mt-nd1 as the mitochondrial control. mtDNA copy number PCR was performed using the Sensifast SYBR Lo-ROX kit (Bioline, USA).”

Results (Page 8; lines 179-184): “To interpret if the decreased expression of mitochondrial transcripts was due to a decrease in mitochondrial number, we did mtDNA copy number analysis on *Lrpprc* homozygous mutants. mtDNA copy number was not observed to be significantly decreased in *Lrpprc^GBT0235/GBT0235^* mutants as compared to wild-type (Figure 4—figure supplement 1). These findings are consistent with data observed in the previous studies that have shown a crucial role for *Lrpprc* in the mitochondrial mRNA stability”

Reviewer #3 (Recommendations for the authors):In this manuscript, the authors aim to create a zebrafish model for Leigh Syndrome French Canadian Type (LSFC) by knocking out the irpprc gene responsible for the disease. In previous works, the authors collected transgenic fish lines that labelled specific proteins via a protein trap approach and identified the Lrpprc knockout fish. The authors analyzed the homozygous mutant phenotypes by RNA-seq, microscopic analyses, electron microscopy, and a fluorescent fatty acid analog. The authors found deficits in liver and muscle development and in liver functions (including lipid metabolism). Further, taking advantage of the feature of the protein trap vector, the orientation of the protein trap vector was reversed and the mutant phenotypes were rescued. Thus, by using a useful model vertebrate zebrafish, the authors successfully established a conditionally controllable LSFC model, which should be useful to understand the function of the Lrpprc gene and the mechanism of the disease.The major strength of this work is (1) detailed descriptions of the mutant phenotypes (lipid accumulation in the liver, mitochondrial morphology defects, and altered lipid metabolism), and (2) successful demonstration of genetic rescue of the mutant phenotypes.The weakness of the work is (1) a mouse model in which the Lrpprc gene was knocked out specifically in the hepatocytes has been described previously, (2) overall the manuscript was rather descriptive and poor in mechanistic aspects of the Lrpprc function and pathology of LSFC, and (3) it was not clearly shown how this genetic model is useful to study the function of Lrpprc and open a way to new therapeutic options as the authors discussed in the Discussion section.1) In abstract, the authors say, "created 1200 transgenic zebrafish strains tagging protein-coding genes". However, in their previous manuscript (Ichino et al. 2020), out of 1200, 204 lines were proved to actually trap proteins. Therefore, this number should be 204.

We thank the reviewer for the comment. In the previous manuscript (Ichino et al. 2020), 1200 new protein trap lines were generated, and the mechanism of expression tied to protein trapping was extensively documented. Over 200 lines out of 1200 were clonally validated and molecularly characterized. We have revised the manuscript to clearly articulate how the numbers we cite correspond to different levels of analyses for the lines in this collection.

We have modified the text and discussed the above findings in the revised manuscript:

Introduction (Page 4; lines 74-76): “Previously, we have reported a compendium of approximately 1200 zebrafish transgenic lines created using the gene-breaking transposon approach out of which 204 were clonally validated and molecularly characterized.”

2) The mechanistic analysis of muscle defects was lacking.

We appreciate the suggestion by the reviewer. In the present manuscript, we have described manifestations that are observed in LSFC patients, one of which is found as muscle defects. In the revised manuscript, we have included a possible explanation of the muscle defects observed in the *Lrpprc* homozygous mutants. Future genetic testing using muscle-specific Cre lines, for example, could be used to separate the potential role of targeting *Lrpprc* in muscle for enhanced therapeutic outcomes. To address this comment and as described in the response to reviewer 1’s comment 1, we were intrigued to see if the muscle phenotype is dependent on the defects in the liver lipid homeostasis.

To see the global effect of liver-specific rescue, we investigated muscle phenotype in the liver rescued *Lrpprc* homozygous larvae, *Tg(fabp10:Cre)/Lrpprc^GBT0235/GBT0235^*. We measured the muscle birefringence of the mutants at 4 days post-fertilization to address the above-mentioned question. The analysis from the muscle studies on mutants revealed that *Lrpprc* homozygous larvae displayed a lower birefringence signal highlighting a disrupted muscle structure/integrity. A similar phenotype was also observed in liver-rescued homozygous animals, where muscle integrity was not observed to be restored (data provided below). These results possibly highlight that the muscle defects do not contribute to the larval lethality that we see in the loss of function embryos and liver rescue wasn’t able to restore the muscle phenotype, despite rescued larvae showing improved survival.

We have modified the text and discussed the above findings in the revised manuscript:

Results (Page 8; lines 194-197): “Assessment for the muscle phenotype in the 4 dpf liver-rescued homozygous mutants displayed a similar decrease in the parameters for muscle integrity such as integrated birefringence density as compared to homozygous mutants (Figure 5—figure supplement 1).”

Discussion (Page 16; lines 421-425): “Interestingly, muscle phenotypes that were displayed by *Lrpprc* homozygous mutants were not mitigated by the liver-specific reversion of the mutagenicity cassette. This not only underscores the specificity of the revertible cassette in a given tissue but also indicates the improved survival and lipid homeostasis observed in rescued animals being at least partially independent of muscle dysfunction.”

Materials and methods (Page 32; lines 694-696): “*Tg(fabp10:Cre)/Lrpprc^GBT0235/+^* were incrossed to obtain *Tg(fabp10:Cre)/Lrpprc^GBT0235/235^*, homozygous mutants only expressing Cre recombinase in the liver.”

3) The mechanism how down-regulated and upregulated genes in the mutants are controlled by Lrpprc was not experimentally analyzed and described.

To understand how the *Lrpprc* deficiency leads to downregulation of mitochondrial encoded transcripts, we established an approach to understand the structure of zebrafish and human *Lrpprc* protein and identify the PPR or RNA binding domains for the same. Having structural insights into this protein will allow us to investigate the homology of the critical functional domains that have been studied in the human orthologue. Using the AI-based structure prediction tool, RoseTTAFold, we were able to predict the structure of the zebrafish Lrprrc protein for the first time. We did annotate the PPR motifs and our analyses revealed that Zebrafish *Lrpprc* shares 51% homology with its human ortholog and has 31 predicted similar PPR domains as compared to 34 PPR domains to its human counterpart, indicating the conserved mitochondrial mRNA binding role of this gene. Primary sequence analysis and structural analysis highlighted the loss of domains important for maintaining the stability of mtRNA transcripts such as RPN7, DUF2517, SEC1, and EAD11 present in the C-terminus region of *LRPPRC* where a single disruptive insertion led to LSFC-associated phenotype in zebrafish. We identified several mitochondrial encoded genes as targets using the FIMO motif search tool of the MEME Suite and matches with the twenty lowest P-values were shown in supplementary tables 1 and 2. The human and zebrafish ortholog do share an overlapping signature of transcripts having binding sites in the respective PPR motifs. The candidates identified from this list were observed to be downregulated in the RNA sequencing datasets and validated by qPCR. Previously, human *LRPPRC* was reported to regulate transcripts coding for complex IV of mitochondrial electron transport chain which was very well aligned to our findings and established our hypothesis that both human and zebrafish *LRPPRC* might have similar mitochondrial targets.

We have modified the text and discussed the above findings under the sections mentioned below in the revised manuscript:

Materials and methods (pages 26-27; lines 535-560): “Structure-based prediction of RNA binding targets for *Lrpprc* protein: To predict the target transcripts of human and zebrafish *Lrpprc*, we first calculated the AI-based models for both human and zebrafish full-length *Lrpprc* proteins using RoseTTAFold. RoseTTAFold is a three-track neural network-based system and has enabled us to predict the structures of the proteins at a high level of accuracy and detail (33). Both the molecular models were computed by using the mforge servers of Mayo Clinic hosted by the University of Illinois Urbana-Champaign (UIUC). The resulting structure files were then rendered with PyMOL (The PyMOL Molecular Graphics System, Version 2.5.2 Schrödinger, LLC). Multiple sequence alignment of different *LRPPRC* sequences from different model organisms was conducted using CLUSTAL Omega v.1.2.4 (52) by taking the default parameters and visualized by using Jalview v.2.11.1.7 (53) software. The homology domain organization was predicted by using the Pfam HMM database search (http://pfam.xfam.org/). PPR motifs were predicted and annotated using the TPRpred (32) server and also by independently assessing the zebrafish and human *Lrpprc* predicted structures.

We used this structural information and the PPR cipher code to identify consensus binding sequences (Figure 2—figure supplement 1 and figure supplement 2). We determined the putative target nucleotide sequence by using the identities of the amino acids at the 5th and 35th positions of each PPR motif of human and zebrafish *Lrpprc* proteins. We used the IUPAC degenerate nucleotide codes to obtain a consensus nucleotide sequence. These consensus sequences were then used to identify potential binding targets of human and zebrafish *Lrpprc* by deploying the FIMO motif scanning program in the MEME Suite (https://meme-suite.org/meme/tools/fimo). The FIMO program searches for the occurrences of known motifs in the sequence databases. The putative nucleotide sequence was then used to search human and zebrafish *Lrpprc* RNA binding sites against the mitochondrial genomes of humans (NC_012920.1) and zebrafish (NC_002333.2) respectively using the FIMO program. The predicted target sites were ranked by P- values calculated by FIMO (54).”

Results (pages 5-6; lines 107-145): “Structure-based analysis reveals *Lrpprc* binding sites in the mitochondrial transcriptome: PPR proteins contain tandem arrays of degenerate ~35-amino acids motifs that fold into a hairpin of antiparallel α-helices (19, 20). PPR proteins bind to their target RNAs in a modular and base-specific fashion (19). The 5th and 35th amino acids of each PPR motif determine the target nucleotide, and the ordering of the repeat motif elements in the respective PPR proteins can function as determinants to assign sequence binding specificity (20). To investigate the functional homology for PPR motifs in *Lrpprc* orthologs from humans and zebrafish, we first wanted to identify and annotate the PPR motifs. Since PPR motifs fold into a distinct structure of antiparallel α-helices, we were interested to interrogate the structure of the *Lrpprc* protein to estimate the occurrence of such helices across its tertiary structure. As no traditional structures for these protein orthologs were available, we derived predicted models using the AI-based protein structure prediction tool, RoseTTAFold. These models were used to identify the PPR motifs throughout the human and zebrafish *Lrpprc* proteins and to evaluate any structural similarities between the orthologs. Interestingly, many of the resulting derived PPR motifs in human *LRPPRC* were observed to be shorter than the canonical 35 amino acid PPR consensus motifs (Figure 2). However, the human and zebrafish *Lrpprc* protein structures revealed that the signature PPR motif helix-turn-helix repeats were packed in a rigid conformation (Figure 2A-B). We manually annotated the occurrence of the hairpin of -helices (PPR motifs) throughout the human and zebrafish *Lrpprc* models and numbered each starting from the N-termini of the proteins. Each hairpin of the -helix is believed to bind to a single nucleotide target. This analysis identified 34 and 31 PPR motifs in the human and zebrafish *Lrpprc* orthologs, respectively (Figure 2—figure supplement 1 and 2).

Analysis of these structures identified a series of standard PPR motifs and their resulting likely RNA-targeting amino acid side chains for sequence-specific interactions. Some PPR motifs of the human and zebrafish *Lrpprc* proteins were also noted to be less regular in form. For example, some motifs did not have obvious typical amino acids at the 5th and 35th positions to confer the nucleotide specificity and were longer or shorter than typical canonical PPR motifs. The resulting inferred consensus sequences (Figure 2—figure supplement 1 and 2) represent candidate RNA binding targets for these proteins.

To predict the potential transcripts bound by *Lrpprc* proteins, we took advantage of the recent development in executing the FIMO program to search the potential targets for several plant PPR proteins (21, 22). The predicted putative targets for human and zebrafish *Lrpprc* (Figure 2—figure supplement 1 and 2) were used to query the complete human and zebrafish mitochondrial transcriptomes, respectively. We identified several mitochondrial-encoded genes as targets using the FIMO motif search tool of the MEME Suite, and the top twenty hits with high confidence scores are listed in supplementary tables 1 and 2 for humans and zebrafish respectively. In addition, the structure of zebrafish *Lrpprc* protein revealed that the C-terminus half, which was deleted in the zebrafish GBT mutant allele, consisted of multiple PPR motifs for RNA-binding (Figure 2C and Figure 2—figure supplement 2).”

Discussion (pages 12-14; lines 298-376): “Zebrafish *Lrpprc* shares 51% homology with its human ortholog and has 31 predicted similar PPR domains indicating the conserved mitochondrial mRNA binding role of this gene. The spatiotemporal analysis of *Lrpprc*-mRFP shows a co-localization with an established mitochondrial marker in the caudal fin region and myocytes, as the truncated fusion proteins retain the mitochondrial targeting sequence maintaining its predicted subcellular localization. We also observed remarkable ubiquitous expression patterns throughout the organism and hotspots in the liver, skeletal muscle, and eye. In addition, increased expression of the fusion protein in certain tissues such as the liver, skeletal muscle, and eyes is indicative of the observation that mitochondrial density is tied to energy-intensive processes like metabolism, movement, and vision. Differential expression data will nevertheless help in expanding our understanding of the different roles of this gene in tissue-specific niches.

Sequence analysis of the predicted zebrafish *Lrpprc* protein revealed the presence of some known homology domains like secretory (SEC)1, RPN7, DUF2517, and EAD11. SEC1 homology domain-containing proteins are known to be involved in vesicular trafficking, synaptic transmission, and general secretion (8, 29). In addition, the human *LRPPRC* protein interacted with various other protein partners like fibronectin through its SEC1 domain (8). The effector-associated domains (EADs) are small, non-catalytic domains and are predicted to be involved in protein-protein interactions (30). The RPN7 domain is one of the regulatory subunits of the 26S proteasome and is known to be required for structural integrity in *Saccharomyces cerevisiae* (31). The region of zebrafish *Lrpprc* from amino acids 966 to 1012 had homology towards the protein of unknown function (DUF2517) domain, which is common to proteobacteria. The function of the DUF2517 homology domain is still unknown. Our in silico structural studies demonstrated that the C-terminus of human *LRPPRC* was well aligned with the C-terminus of zebrafish *Lrpprc* (Figure 2—figure supplement 3), strengthening the hypothesis that the zebrafish *Lrpprc* is playing a critical role in cell metabolism as discussed above. Additionally, the multiple sequence alignment of *LRPPRC* showed that the C-terminus region of this protein was conserved between several model organisms (Figure 2—figure supplement 3). Primary sequence and structural analyses highlighted the loss of domains important for maintaining the stability of mtRNA transcripts such as RPN7, DUF2517, SEC1, and EAD11 present in the C-terminus region of *LRPPRC* where a protein truncation led to LSFC-associated phenotype in zebrafish.

To understand the phenotypic and metabolic findings in the *Lrpprc*-deficient zebrafish model, we also analyzed the potential RNA binding targets of the *Lrpprc* protein. *Lrpprc* contains multiple PPR motifs, and these functional domains determine the target RNA binding sites. We first annotated the PPR motifs for the *Lrpprc* protein for zebrafish and human orthologs to explore RNA binding sites. Preliminary analysis using the TPRpred server identified 22 PPR motifs for both human and zebrafish *Lrpprc* proteins (22). TPRpred is a profile-based method that uses a P-value-dependent score offset and exploits the tendency of repeats to occur in tandem (32). This tool not only predicts the Tetratrico Peptide Repeat (TPR) motifs, but also the related PPRs and SEL1-like repeats. However, a study by Spåhr and colleagues had predicted 33 PPR motifs for the human *LRPPRC* protein (7) which was more than what was predicted by the TPRpred (32) server. Additionally, we evaluated the secondary structure containing the *Lrpprc* proteins by PSIPRED server, we found that both humans and zebrafish are all α-helical and may contain more hairpin of -helices than what TPRpred predicted. To address this discrepancy and to conduct a meticulous study to systematically identify PPR motifs in human and zebrafish *Lrpprc*, we employed a deep-learning-based structure modeling tool, RoseTTAFold (33). Using RoseTTAFold we designed a workflow to predict 31 and 34 PPR motifs for the zebrafish and human orthologs respectively of *Lrpprc* proteins.

We investigated the sequences of the identified PPR motifs in each of the orthologs and deciphered the RNA recognition residues, which are typically found at the 5th and 35th positions in each PPR motif. These two amino acids play a major role in specifying the target nucleotide and constitute a code for nucleotide recognition. The potential target recognition codes for human and zebrafish *Lrpprc* PPR motifs were predicted to exhibit a range of nucleotide targets (Figure 2—figure supplement 1 and 2). We used the FIMO search algorithm to predict the potential binding sites for human and zebrafish *Lrpprc* proteins. In silico RNA-binding analysis revealed that transcripts coding for proteins involved in the complex IV activity such as mt-co1 and mt-co2 were among the hits with a high confidence score. Apart from complex IV genes, transcripts belonging to complex I and complex V were also identified to be recognized by the PPR motifs of the *Lrpprc* protein. There was a consensus for the majority of the target transcripts for the zebrafish and human orthologs indicating a conserved function of *Lrpprc* in the maintenance of the mitochondrial genome both in humans and zebrafish. Previously, human *LRPPRC* was reported to regulate transcripts coding for complex IV of mitochondrial electron transport chain (34, 35) which was very well aligned with these findings and supports the hypothesis that both human and zebrafish *LRPPRC* likely have similar mitochondrial targets. To the best of our knowledge, this is the first such analysis that has been performed and validated for PPR-based proteins.

Previously published reports have highlighted the role of PPR-containing proteins in mediating mRNA stability and translational activation (36, 37). PPR proteins have been shown to stabilize the RNA structure by blocking the access to exonucleolytic cleavage upstream and downstream of their binding sites in a way acting as protein caps at the 5’ and 3’ ends of the processed transcripts (36). It also confers a site-specific remodeling activity by sequestering the target RNA by preventing it from binding to the translation initiation region (36). Another functionality contributed by this class of proteins is aiding in the inhibition of uridylation of A-tailed transcripts, thereby preventing the degradation of mRNAs (37). These roles highlight the role(s) of PPR motif-containing proteins in the stability of the mitochondrially encoded transcriptome.

Upon further investigation using independent transcriptomics sequencing and targeted mitochondrial RNA qPCR, it was observed that downregulated mtDNA genes in the *Lrpprc* homozygous zebrafish mutants overlapped with those predicted by the above algorithm. These results possibly explain the reduced expression of mitochondrial RNAs in the homozygous mutants, underscoring the role of *Lrpprc* as loss of this protein impedes the binding of this protein, therefore leading to mRNA instability and degradation.”

Discussion (pages 14-15; lines 384-392): “*LRPPRC* protein in mitochondria forms a complex with SRA stem-loop interacting RNA binding protein (SLIRP) protein in the mitochondrial matrix. The absence of *LRPPRC* leads to oligoadenylation of the transcripts. *LRPPRC*-SLIRP is also observed to relax RNA structure, making the 5' end of the mRNA accessible for the mitochondrial ribosome to initiate the translation (38). The mechanistic insights in our zebrafish *in vivo* model from the structure of *Lrpprc* protein and target sites of the predicted PPR domains as described earlier correlate with observations that have been well established in patient-derived cell culture model displaying the altered signature of mitochondrial transcripts (11, 23).”

Supplementary table 2 (page 49): “Prediction of Human *LRPPRC* binding sites within the human mitochondrial genome (NC_012920.1). The top twenty matches among both strands of the complete mitochondria genome are shown. The P-values were calculated with the FIMO program and were used for ranking.”

Supplementary table 2 (page 50): “Prediction of zebrafish *Lrpprc* binding sites within the zebrafish mitochondrial genome (NC_002333.2). The top twenty matches among both strands of the complete mitochondria genome are shown. The P-values were calculated with the FIMO program and were used for ranking.”

4) How was the mitochondria morphology altered was not shown.

We appreciate the comment by the reviewer. Mitochondrial morphology is a classical hallmark of underlying mitochondrial dysfunction. Mitochondria can undergo changes in shape accompanied by fusion and fission events which may involve remodeling at cristae junctions that dynamically appose and separate from each other. Under TEM, hepatic mitochondria had altered morphology with observed collapsed cristae in the homozygous mutants, potentially impairing mitochondrial function. From the transcript studies and motif-based target prediction, we did observe a decreased expression of *mt-atp6* and *mt-atp8* transcripts due to *Lrpprc* deficiency. Proteins coded by these mRNAs constitute the complex V of the electron transport chain. It has been reported that complex V plays an essential role in defining the cristae morphology specifically in the areas of high membrane curvature at cristae ridges. Addressing these comments, we have discussed this observation in the revised manuscript wherein we have highlighted how the abolishment of domains critical for mtRNA binding in *Lrpprc* can influence the stability of mitochondrial-encoded transcripts. Though, detailed experiments elucidating the molecular mechanisms of cristae remodeling are required to delve more into these hypotheses.

We have modified the text and discussed the above findings under the sections mentioned below in the revised manuscript:

Results (Page 11; lines 280-281): “Liver-specific rescue in the homozygous mutant background mostly reverted the mitochondrial morphology to an extent (Figure 7F).”

Discussion (Page 16; lines 439-448): “Mitochondrial morphology is the classical hallmark of underlying mitochondrial dysfunction. Mitochondria can undergo changes in shape accompanied by fusion and fission events which may involve remodeling at cristae junctions that dynamically appose and separate from each other (42, 43). Under TEM, hepatic mitochondria had an altered morphology in the homozygous mutants, potentially impairing mitochondrial function. From the transcript studies and motif-based target prediction, we did observe a decreased expression of mt-atp6 and mt-atp8 transcripts due to *Lrpprc* deficiency. Proteins coded by these mRNAs constitute the complex V of the electron transport chain. It has been reported that complex V plays an essential role in defining the cristae morphology specifically in the areas of high membrane curvature at cristae ridges (10).”

Discussion (Pages 18-19; lines 503-512): “Mitigation of abnormal mitochondrial structure was also observed in the hepatocytes of the rescued larvae. Although we did observe that rescue was not completely reversible in the liver-specific Cre homozygous mutants. There may be several factors that can contribute to this observation such as incomplete penetrance or rescue of the mutant allele. As cristae are dynamic structures of mitochondria whose morphology is altered under various bioenergetics and physiological conditions (42, 43), reversion of mutagenicity cassette in the hepatocytes is not sufficient to correct cristae alterations that appeared to be aggravated in the *Lrpprc* homozygous mutants. A possible explanation for the restoration in the morphology may be attributed to the restoration of levels of mitochondrial transcripts and thus restoring mitochondrial-mediated liver functions such as β-oxidation and fatty acid elongation.”

5) The mechanism to control lipid metabolism was totally unknown.

We do appreciate the comment by the reviewer and during the revision of the manuscript, we did try to explore the retention of lipids in the homozygous mutants. As described in the previous version of the manuscript that oil red o staining in the *Lrpprc*^GBT0235/GBT0235^ mutants revealed an abnormal distribution of lipids with increased lipid accumulation in the livers of these animals, suggesting the observed darkened liver phenotype could be caused by a change in the density of the stored lipids. By coupling the feeding of a fluorescent fatty acid analog with high-performance liquid chromatography methods, triglycerides and diglycerides, the major components of nonpolar lipids along with cholesterol esters were observed to be higher in null mutants, highlighting the defect in fatty acid metabolism underscored by the accumulation of lipids observed by Oil Red O staining. These lipid metabolism defects were restored in liver-rescued zebrafish embryos (Tg(fabp10: Cre)/*Lrpprc*^GBT0235/GBT0235^), shedding light on the liver as a major metabolic organ during the pathogenesis of LSFC.

To explore further explore the retention and sequestration of these synthesized triglycerides in the mutants, we fed them with lipid-rich egg yolk (Otis, J. P., Farber, S. A. High-fat Feeding Paradigm for Larval Zebrafish: Feeding, Live Imaging, and Quantification of Food Intake, JoVE 2016).

Fed larvae were fixed the larvae at different time points post-feeding, i.e. 0 hours post-feeding and 13 hours post-feeding. Unfed embryos were selected as the control group. Oil Red O staining was performed on fixed embryos and was observed specifically for the signal in the intestine. This would be intriguing to check for any lipid accumulation in the intestine suggesting non-cell-autonomous rescue. However, we didn’t see any observable difference in the lipid sequestration in the gut and intestine in the *Lrpprc* homozygous siblings as compared to their rescued mutants and wild-type siblings.

We do acknowledge that there is an interesting aspect of lipid biology that needs to be explored further in *Lrpprc* mutants. Findings from this study in the LSFC animal models shed unique insights towards *in vivo* lipid biochemistry to explore novel lipid metabolism mechanistic insights in the homozygous and liver-rescued animals in the future. However, those experiments are in a nascent stage and are at present beyond the scope of the aims of the present manuscript. We do believe findings from future studies stemming from these observations would potentially add new paradigms to the role of liver and lipid management in the progression of LSFC.

Additional study:

In addition to the useful suggestions and comments from the reviewers, we did explore an additional but clinically important phenotype on the LSFC zebrafish model by measuring the lactate levels. Details of these experiments and modifications in the revised manuscript have been described below.

1. Measurement of lactate:

The current diagnosis of LSFC hinges on measuring lactate levels in the blood and brain, COX activity in patient fibroblasts, and sequencing of the *LRPPRC* gene for mutations. Homozygous mutants were interrogated to assess the levels of lactate, a clinical mitochondrial marker that is observed to be altered in patients presenting with LSFC symptoms. Analysis of zebrafish lysate using gas chromatography/mass spectrometry (GC/MS) revealed an abnormal threefold increase of lactate in *Lrpprc^GBT0235/GBT0235^* mutants.

Methods (page 31; lines 680-691): “Lactate measurement in zebrafish embryos- *Lrpprc^GBT0235/+^* zebrafish were incrossed and sorted as per genotype. Seven dpf zebrafish larvae were pooled together and lysed in 100 µL of 1X PBS. 100 µL of lysate was spiked with a mixture of labeled internal standards following PFB-oximation of keto acids. The samples were acidified with 6N HCl and NaCl and extracted into ethyl acetate. After evaporation, the dry residue was silylated with N,O,-bis-(trimethylsilyl) trifluoroacetamide with 1% trimethylchlorosilane (BSTFA+TMCS). Subsequently, the reaction mixture was analyzed by capillary gas chromatography/mass spectrometry (GC/MS) (GC: Agilent 7890B Gas Chromatograph; mass spectrometer is an Agilent 5977A Mass Spectrometer, EI Source, and ionization) using selected ion monitoring (SIM) with positive electron impact ionization (EI) and stable isotope dilution. Peaks were acquired using Agilent Masshunter GC/MS Acquisition 10.0.368 and the software used for quantitation was Agilent Quantitative Analysis Masshunter Version 10.0.”

Results (page 8; lines 184-187): “Homozygous mutants were also interrogated to assess the levels of lactate, a clinical mitochondrial marker that is observed to be altered in patients presenting with LSFC symptoms. Analysis of zebrafish lysate using gas chromatography/mass spectrometry (GC/MS) revealed increased levels of lactate in *Lrpprc^GBT0235/GBT0235^* mutants (Figure 4C).”